# Phytochemicals and Biological Activities of Walnut Septum: A Systematic Review

**DOI:** 10.3390/antiox12030604

**Published:** 2023-03-01

**Authors:** Letiția Mateș, Marius Emil Rusu, Daniela-Saveta Popa

**Affiliations:** 1Department of Toxicology, Faculty of Pharmacy, Iuliu Hatieganu University of Medicine and Pharmacy, 8 Victor Babes, 400012 Cluj-Napoca, Romania; 2Department of Pharmaceutical Technology and Biopharmaceutics, Faculty of Pharmacy, Iuliu Hatieganu University of Medicine and Pharmacy, 8 Victor Babes, 400012 Cluj-Napoca, Romania

**Keywords:** walnut septum, by-products, diaphragma juglandis fructus, antioxidants, anti-aging, bioactive compounds, biological activity, in vitro, in vivo, toxicity

## Abstract

In the last few decades, scientific evidence has stressed the importance of plants in the prevention and/or supportive treatment of a plethora of diseases, many of them chronic, age-associated disorders. *Juglans regia* L. is a traditional plant that has been integrated into traditional medicine since ancient times. Due to the presence of biologically active compounds, walnut was used in the treatment of various maladies. Recently, investigations have focused on the walnut by-products and waste products, with research on their valuable constituents and active properties. Among these secondary products, walnut septum was analyzed in several studies, its phytochemical profile described, and some of the biological activities examined. However, compared to other walnut by-products, no comprehensive review to gather all the pertinent scientific knowledge was found in the literature. Therefore, the aim of this study was to critically assess the information furnished by peer-reviewed articles regarding the walnut septum chemical composition and the related biological activities, including antioxidant activities, anti-inflammatory effects, antimicrobial properties, antidiabetic activities, anti-tumor properties, and anti-aging potential. In conclusion, as these preclinical studies showed that walnut septum metabolites were responsible for a wide range of preventive and therapeutic uses, further research should confirm the beneficial outcomes in clinical trials.

## 1. Introduction

Plant medicine is increasingly used by people due to their interest in complex natural products that are as compatible as possible with the human organism and have as few side effects and adverse reactions as possible. Therefore, the use of herbal remedies must be based on scientific evidence, including recent biological screening methods in combination with analytical approaches [1]. Diets rich in fruits, vegetables, and nuts are commonly recommended for their health benefits due to their concentrations of vitamins, minerals, phytochemicals, and dietary fiber [2].

Nuts and seeds, along with whole grains, vegetables, fruits, and legumes, are essential components of globally recommended healthy diet patterns, such as the Mediterranean diet [3] and the Dietary Approaches to Stop Hypertension (DASH) diet [4]. Tree nuts and peanuts are nutrient-dense foods with complex phytochemical matrices [5]. They are rich in unsaturated fatty acids, folic acid, vitamin E, potassium, calcium, magnesium, phytosterols, polyphenols, and fiber [5].

Among the tree nuts, walnut (*Juglans regia* L.) is a valuable cultivated tree, widely appreciated for its fruits [6]. The walnut seed, or kernel, is rich in lipids, predominantly unsaturated fatty acids (MUFA and PUFA), phytosterols, tocopherols, proteins, minerals, and a number of antioxidant chemicals, such as phenolic compounds [7,8,9]. It has an important medicinal value and health function in preventing and alleviating cardiovascular diseases, diabetes, and obesity [7]. Some walnut by-products have been studied and shown to be rich in bioactive constituents [10]. Walnut leaves, husks, and shells contain key phenolic compounds and are often used in traditional medicine to treat different ailments [6,11]. Walnut skin, or pellicle, is the thin, brown coating that prevents oxidation and microbial contamination of the seed [11]. It is walnut’s main source of polyphenols with powerful antioxidant and anti-inflammatory capacities [6].

Walnut septum (WS), a by-product that is frequently discarded as waste during walnut processing, has a phytochemical profile that defines the therapeutic potential. Recent studies exposed that WS contains many bioactive phytochemicals, with demonstrated both in vitro and in vivo biological activities including anti-aging potential [12,13,14].

The aim of this study was to critically examine and synthesize available data on the chemical composition and biological activities of WS. To the best of our knowledge, this is the first exhaustive systematic review of this plant matrix, and it reveals that WS metabolites could be responsible for many preventive and therapeutic actions.

## 2. Search Methodology

The current systematic review was performed following the PRISMA criteria guidelines [15]. The registration code is INPLASY202320075, with DOI 10.37766/inplasy2023.2.0075, https://inplasy.com/?s=INPLASY202320075 (accessed on 16 February 2023).

### 2.1. Eligibility Criteria

Our systematic review included studies performed on walnut septum plant material or walnut septum with the following objectives: (1) identification and/or quantification of phytochemical compounds; (2) examination of the biological activity of identified compounds via in vivo and in vitro testing; and (3) English and French publications. We excluded: (1) abstracts, narrative reviews, comments, opinions, methodological papers, editorials, letters, observational studies, conference abstracts, or any other publications lacking primary data and/or explicit method explanations; (2) publications with full text not available and the corresponding author could not be contacted; and (3) duplicate studies or databases.

### 2.2. Information Sources

We performed a systematic literature search in PubMed, EMBASE, and ClinicalTrials.gov databases for studies describing the phytochemical profile and biological activity of walnut septum from the inception of each database through to 31 January 2023. The literature search had no language constraint. To ensure thorough research, the bibliographies of the included studies and current reviews were also screened.

### 2.3. Search Strategy

To search the databases, we used a combination of free-text words, along with their synonyms, singular and plural forms, thesaurus words (Medical Subject Headings for PubMed: (“juglans” [MeSH Terms] OR “juglans” [All Fields] OR “walnut” [All Fields] OR “walnuts” [All Fields] OR “juglans regia” [All Fields] OR “juglans nigra” [All Fields]) AND (“septum” [All Fields] OR “septa” [All Fields] OR “diaphragma juglandis” [All Fields] OR “diaphragma juglandis fructus” [All Fields] OR “internal septum” [All Fields]), and Emtree for EMBASE: (‘juglans’/exp OR ‘juglans’ OR ‘walnut’ OR ‘walnuts’ OR ‘juglans regia’ OR ‘juglans nigra’ OR ‘english walnut’ OR ‘persian walnut’) AND (‘walnut septum’ OR (‘walnut’ AND ‘septum’) OR ‘diaphragma juglandis’ OR (‘diaphragma’ AND ‘juglandis’) OR ‘walnut internal septum’ OR (‘diaphragma’ AND ‘internal’ AND ‘septum’)).

### 2.4. Selection Process

All three authors independently reviewed the titles and abstracts of relevant journal publications. Then, the full texts of the documents that appeared to meet the selection criteria were collected for additional screening. Each full text was independently evaluated by the same researcher. In the event of discord, the research was discussed until a consensus was achieved. In cases where there were numerous publications from the same trial, only the most relevant or useful article was selected.

### 2.5. Data Items

Data regarding the outcomes were extracted in a spreadsheet Microsoft (Microsoft Office 365, MS, Redmond, WA, USA) Excel file contained the following data: materials/type of extract, phytochemical composition studies, in vitro studies/ biological systems analysis methods, and in vivo studies/animal models. Furthermore, data regarding study characteristics were extracted in a spreadsheet file: country, study design, study purpose, and study outcomes.

Other investigators than those who extracted the initial full-text articles rechecked the extracted data.

A total of 56 articles were considered from the systematic search and review of relevant reference lists. After applying exclusion criteria, 28 articles were included in the systematic review. The procedure of study inclusion and exclusion is shown in Figure 1. The characteristics of included studies are revealed in Table 1.

## 3. Phytochemical Composition

The phytochemical composition of walnut septum was examined in 13 out of the 28 studies selected for this review. Five of these provided quantitative determination results that could be analyzed and compared in our study [24,27,32,37,40] (Table 2).

The reviewed investigations determined the phytochemical profile focusing on the examination of the phenolic compounds, lipids, carbohydrates, amino acids, and mineral compositions.

Eleven studies examined phenolic components, including phenolic acids and their esters, flavonoids, tannins, lignans, and phenylpropanoids. Fatty acids, phytosterols, tocopherols, terpenoids, and carotenoids were among the lipid constituents identified, while the monosaccharides and polysaccharides were the main carbohydrates of walnut septum. Other compound types including coumarins, quinones, and alkaloids, known to be present in various matrices of *Juglans regia* L., are less extensively explored and discussed in these articles. However, the representative data for these compounds, extracted from the selected studies, are presented in this review.

### 3.1. Phenolic Compounds

The research that identified phenolic compounds in the walnut septum was summarized, according to their methodology, in Table 2. The chemical structures of quantitatively representative phenolic compounds found in walnut septum were illustrated in Figure 2.

It is widely recognized that the extraction yield and the conservation of the structural and biological properties of valuable components are highly dependent on the extraction method applied [41].

Most of the reviewed studies reported the usage of WS in powder form, following the preparatory grinding of plant material. The extractions were conducted in ethanol [20,22,38,40] or methanol [26,27,32] at various concentrations, as well as in solvent mixes such as acetone:water [24], acetone:acetonitrile [25], and acetone:cyclohexane [27]. The extraction methods were generally based on ultra-turrax homogenization and stirring [24], vibrating [25], ultrasonic extraction [26,27,37], condensation and reflux, or enzyme-assisted extraction [32], then centrifugation, solvent evaporation, and lyophilization. A number of studies described preserving the lyophilized material at −20 °C [25,26,32], while others stored it at room temperature [24] (Table 2).

The reviewed studies used various chromatographic techniques for identifying phenolic compounds: liquid chromatography-mass spectrometry (LC-MS) [24], liquid chromatography with tandem mass spectrometry (LC-MS/MS) [32], high-performance liquid chromatography (HPLC) [20,22,26,37,38], or ultra-performance liquid chromatography (UPLC) differently equipped: UPLC-MS (17), UHPLC-Q-Orbitrap coupled with high-resolution mass spectrometry HRMS [27], or UPLC-Q-Exactive Orbitrap MS [40].

In order to quantify the detected compounds, the majority of the studies utilized electrospray ionization mass spectrometry (ESI-MS) with positive and/or negative mode, nuclear magnetic resonance (NMR): 1H-NMR, 13C-NMR, and ultraviolet (UV) spectroscopy (280–360 nm).

The phenolic compounds identified in WS belong to the following classes: phenolic acids (hydroxybenzoic, hydroxycinnamic, hydroxyphenylacetic acids), and some of their esters: flavonoids (flavones, flavonols, chalcones, anthocyanins, flavanones, flavanols), tannins (gallotannins, ellagitannins, complex tannins), lignans (cyclolignans, monoepoxylignan, benzofuran lignan), phenylpropanoids (phenylpropionic acids, phenylpropanols), as well as other classes (hydroxybenzaldehydes).

In the selected studies, the following hydroxybenzoic acids were analyzed: benzoic acid [40], p-hydroxybenzoic acid [22,32,40], 4-hydroxybenzoic acid, 3,4-hydroxybenzoic acid [40], p-coumaroylquinic acid [26,32], ellagic acid [26,27,32], gallic acid [22,24,25,26,27,40] gentisic acid [24], protocatechuic acid [22,24,27,32], syringic acid [24,32,40], and vanillic acid [22,24,25,40] (Table 2).

In the quantitative analyses, Liu R. et al. [27] obtained significant amounts of ellagic acid with a range of 518.38 to 1733.64 μg/g dw for the ten batches investigated. It is noteworthy that the concentration of ellagic acid was comparable to or even higher than that of other plant sources [27]. Similarly, Zhang et al. [32] reported the highest amount of ellagic acid (351.92 g/g dw) for methanolic condensation reflux extraction (ME) for all three tested methods.

Gallic acid was quantitatively assessed in three investigations [24,27,40] with comparable results. In one research [24], gallic acid was found in a significantly higher concentration (79.58 µg/g dw) compared to other detected phenolic acids (protocatechuic, syringic, and vanillic acids). However, this concentration was lower than that determined by Liu R. et al. [27]. The highest quantity of gallic acid (272.52 μg/g dry weight) was encountered in the study of Chen et al. [40].

Protocatechuic acid was found in substantial amounts (a range of 44.28–154.04 μg/g dw in ten samples) compared to the other hydroxybenzoic acid compounds investigated [27].

Hydroxycinnamic acids detected and described were caffeic acid, chlorogenic acid, neochlorogenic acid, caftaric acid, ferulic acid and its isomer, p-coumaric acid, and sinapic acid (Table 2). Among them, the isomer of ferulic acid has the highest concentration (87.96 µg/g dw) in ME [32].

Flavonoids are low-molecular-weight secondary metabolites [42]. The flavonoids identified in this research included flavones (apigenin, luteolin, and vitexin), flavonols (isorhamnetin, kaempferol, dihydrokaempferol, myricetin, myricetol, dihydrokaempferol, quercetin, avicularin, hyperoside, rutin, and various quercetin glycosides, quercitrin, taxifolin, taxifolin-3-O-arabinofuranoside, taxifolin 3-rhamnoside), chalcones (naringenin chalcone), anthocyanins (B-type procyanidin dimer, trimer, tetramer, pentamer, hexamer, and their isomers), flavanones (naringenin 7-O-β-D-glucopyranoside, sakuranetin 5-O-β-D xylopyranoside, (2R)-eriodictyol-5-O-β-D-glucoside), flavanols ((−)-epicatechin, (+)-catechin, (−)- epicatechin gallate).

Fruits and vegetables, herbal tea, and red wine are rich in flavonols, which give them a spectrum of colors from white to yellow. The most prevalent flavonol found in plant sources was quercetin [43]. Quercetin [32,40], isoquercetin [24,40], quercitrin [24,40], taxifolin-3-O-arabinofuranoside [27], catechin [24,27,40], epicatechin [40], and epicatechin gallate [27] were detected in walnut septum in significant quantities compared to other components. Among all the assessments, [40] found the highest values for these compounds: quercetin (54.10 μg/g dw), isoquercetin (399.00 μg/g dw), quercitrin (6816.18 μg/g dw), catechin (9989.16 μg/g dw), and epicatechin (362.10 μg/g dw) (Table 2).

Gallotannins, such as monogalloyl-glucose, digalloyl-glucose, trigalloyl-glucose, and tetragalloyl-glucose [32], as well as their isomers [26], methyl galloyl hexoside, and caffeoyl glucopyranose isomer [32], were detected in the selected studies for this review. Although in very small amounts compared to the other phenolic compounds, monogalloyl-glucose (4.87 μg/g dw, EE) and trigalloyl glucose (4.90 μg/g dw, ME) stood out for their higher amounts between the investigated tannins in the study of Zhang et al. [32].

Elagitanins were identified as ellagic acid hexoside, elagic acid pentoside, and isomers, bis-HHDP glucose [26], HHDP-glucose isomer, and galloyl-HHDDP glucose [26,32]. Quantitatively, distinct compounds were the HHDP-glucose isomer (6.75 μg/g dw, EE) and galloyl-HHDP glucose (5.77 μg/g dw, ME) [32].

The investigation performed by Tan et al. 2022 (a) [38] also reported cyclolignans (5-methoxy-(+)-isolariciresinol), monoepoxylignans (erythro-guaiacyl-glycerol-β-O-4′-(5′)-methoxylariciresinol), rhoiptelol B, and benzofuran lignans (dihydrodehydodiconiferyl alcohol). However, these compounds were not quantified. Certain phenylpropanoic compounds, including 1,6-di-O-(E)-coumaroyl-D-glucopyranoside (phenylpropionic acid) and various phenylpropanols, among them erythro-(7S,8R)-guaiacyl-glycerol-O-4′-dihydroconiferyl ether and rosalaevin B were also identified in the same study.

### 3.2. Lipidic Compounds

The studies that identified lipidic compounds in the walnut septum, along with their methodology, are listed in Table 3.

The lipidic molecules were identified in three of the thirteen investigations that aimed to define the phytochemical composition of walnut septum [12,24,25] and include fatty acids [25], phytosterols [24], and tocopherols [12].

In the study of Hu Q. et al. (2019) [25], the total amount of reported saturated fatty acids (SFAs) was 1099.10 mg/kg dw, with the highest quantity in palmitic acid (C16:0). The proportion of polyunsaturated fatty acids (PUFAs) in WS was 63.5%, with a prevalence of linoleic acid (18:2) and oleic acid (18:1 cis-9). Small amounts of other PUFAs including eicosapentaenoic acid (20:5, n-3), arachidonic acid (20:4, n-6), and nervonic acid (24:1, n9) were also found in WS.

According to the quantitative analysis of the phytosterols beta-sitosterol was found to be ten times more abundant (31,018.16 µg/g dw) than campesterol in the analyzed sample of WS [24]. The identified quantities of tocopherols were significantly lower than those of phytosterols, with a predominance of α-tocopherol [24].

### 3.3. Carbohydrate Compounds

Of the 13 analyzed studies, only two [23,25] reported the evaluation of carbohydrates (monosaccharides and polysaccharides) in WS, of which only one [25] provided information on their quantity. The extracted data were systematized in Table 4, and the structures of the representative compounds (trehalose and xylose) were presented in the Figure 3.

### 3.4. Other Compounds

This category consists of other compounds identified in WS, which were presented in Table 5.

Apart from the previously presented compounds, sesquiterpenoids quinones, coumarins, amino acids, and minerals were also identified in WS.

The majority of the sesquiterpenoid compounds were megastigmanes, including blumenol B [22,25,38], blumenol C glucosides [22,38], and diamegastigmanes (A, B, and C) [38]. This class also includes taraxasteranes (Juglansin A, B, C, and D) [38]. In the study conducted by Hu et al. [25], glutamate and lysine were quantitatively highlighted among the amino acids identified in WS, whereas Ca, K, and Mg were the most abundant minerals (Table 5).

## 4. Biological Activities

Table 6 summarizes the studies reported in the scientific literature that highlight the biological activities of walnut septum extracts or compounds isolated from walnut septum.

### 4.1. Antioxidant Activities

The overproduction of free radicals, such as ROS and RNS, could generate oxidative impairment to biomolecules and consequently lead to many chronic ailments including cardiometabolic disorders, neurodegenerative diseases, and cancers [45]. ROS form in mammalian cells and have distinct effects on oxidative stress, chronic inflammation, and biological aging. Therefore, controlling and targeting ROS sources and overproduction are measures involved in antioxidant, anti-inflammatory, or healthy aging therapeutics [46].

Antioxidants are important molecules and play key roles in food industry against oxidative deterioration of products and against oxidative stress-mediated pathological processes in the body. Several tests were used to determine the antioxidant activity of extracts of walnut septum. The most common methods were DPPH and ABTS, discoloration assays based on electron or hydrogen atom donation, and the ferric reducing antioxidant power (FRAP) assay, a color-formation reaction applicable for both in vitro and in vivo experiments that is based on electron transfer rather than hydrogen atom transfer [47].

Various studies reported the antioxidant activities of phytochemicals in walnut matrices [6]. One of our previous experiments showed good antioxidant activity via ABTS assay for the 75% and 50% acetone septum extracts at 174.28 ± 8.22 and 168.62 ± 9.68 mg TE/g dw septum, respectively, both samples obtained by the UTE method [24]. The best DPPH radical scavenging activity was expressed by an equal acetone/water septum extract [24]. The two antioxidant activities were positively influenced by the number of hydroxyls present in the aromatic ring of the extract phenolics.

Similarly, Zhang et al. revealed that the septum antioxidant capacity increased in a phenolic concentration-dependent manner. Moreover, the antioxidant activity was also affected by the extraction method applied. Thus, septum extracts obtained using methanolic condensation reflux extraction were more active compared to extracts derived through enzyme-assisted (EAE) and ultrasonic wave-assisted extraction (UWAE) methods [32].

The DPPH and FRAP test results of Mehdizadeh et al. demonstrated high antioxidant capability of the septum hydroalcohol extracts compared to butylated hydroxytoluene (BHT) [28].

Hu et al. analyzed the antioxidative effects of two extracts, water and alcohol septum samples obtained using hot water (1:20 *w*/*v*) at 85 °C and 70% ethanol at room temperature, respectively [35]. Both extracts exposed significant HepG2 cell protective capacities and were capable of scavenging DPPH and ABTS in dose-dependent manners [35].

As the major ROS contributor in cells, H_2_O_2_ is considered relevant to endogenous oxidative stress. Treatment with septum extract significantly ameliorated the oxidative damage in a concentration-dependent manner in human hepatic L02 cells exposed to H_2_O_2_ and the polysaccharides found in septum were found to be the active molecules [29]. These saccharides, mostly glucose, followed by galactose and arabinose, and trace amounts of xylose and mannose, indicated that they had antioxidant and bacteriostatic activities [21].

The antioxidant effect of the WSE was evaluated on cancerous (A549, T47D-KBluc, and MCF-7) and normal (human gingival fibroblasts (HGF)) cell lines. In the non-stimulated conditions, as well as the H_2_O_2_-stimulated conditions, the septum extracts decreased the quantity of ROS in a dose-dependent manner [12]. Using the DCFH-DA assay, a dose-dependent and statistically significant decrease of ROS in lung tissues was also noticed, exposing the ability of WSE to mitigate oxidative stress [36].

Tocopherols, key lipophilic radical-scavenging antioxidants, could interrupt the lipid peroxidation cycle. In addition to their direct action against ROS, quercetin glycosides and tocopherols revealed an indirect antioxidant potential by activating the Nrf2/ARE pathway, thus initiating the synthesis of cellular antioxidants including [12]. Other antioxidant biomolecules from septum are the ellagitannins and ellagic acid that could modulate oxidative stress and inflammatory actions via urolithins, their gut microbiota metabolites [48].

The antioxidant activity of septum extracts was also researched on animal aging models, both in old rats and in D-gal-aged young rats [31]. Measured by the TEAC assay, the antioxidant activity in the brain of old Wistar rats significantly increased after repeated (8 weeks) septum extract intake compared to control (9.585 ± 0.287 mg TE/100 g, *p* < 0.05). The septum treatment also improved, but was not statistically significant, the antioxidant activity in the liver when measured by the DPPH assay. Moreover, several oxidative stress biomarkers were lowered after repeated (8 weeks) septum extract intake in old rats. In the liver and brain, ROS, AGE, NO, and MDA levels were reduced after septum treatment but only the liver MDA level was statistically decreased.

However, all these parameters were significantly lowered in the analyzed organs in young animals treated with D-gal to induce aging. The trend was also noticed for acetylcholinesterase (AChE) levels in the brains of both age-groups. Although, as mentioned above, the ROS, AGE, NO, and MDA biomarkers were not significantly modified in old rats after 8 weeks of septum extract treatment, it could be possible that longer time periods of supplementation could statistically impact these indices. The levels of glutathione, a key endogenous antioxidant molecule related to cellular detoxification and redox homeostasis, were increased, although not significantly, after septum extract consumption in this in vivo experiment [31].

In a subacute toxicity study, elevations in the serum, liver, and kidney MDA levels after treatment with a septum methanol extract were noticed, but no severe subacute toxicity was observed for doses of 1000 mg/kg bw after 4 weeks of repeated administration [19].

Recently, Chen et al. evaluated the antioxidant effects of WSE on refined soybean oil during deep frying [40]. The septum extract effectively delayed the thermal oxidation of unsaturated triglycerides and inhibited the formation of potentially toxic components, such as oxidized triglyceride monomers and polymers, or toxic aldehydes. These outcomes could be due to the antioxidant capabilities and synergistic activities of the 31 polyphenols determined in the septum extract, the main compounds being catechin, quercitrin, taxifolin, quercetin 3-β-d-glucoside, epicatechin, gallic acid, and 3,4-dihydroxybenzoic acid [40].

### 4.2. Anti-Inflammatory Effects

The anti-inflammatory effects of fourteen isolated molecules from septum were evaluated by an in vitro model of lipopolysaccharide (LPS)-stimulated murine RAW 264.7 macrophages. Gallic acid, ethyl gallate, and (+)-dehydrovomifoliol were major components that showed potent inhibitory activity on the nitric oxide production (suppression of 70% of NO production) [22]. However, it was highlighted before that relying only on RAW 264.7 cells to assess immune reactivity might not furnish an extensive picture of the bioactive properties of the investigated matrix [49].

Additionally, the ability of WSE to inhibit inflammation in LPS-stimulated HGF cells was evaluated. The results demonstrated a decrease in the levels of interleukin-6 (IL-6), interleukin-8 (IL-8), and interleukin-1 β (IL-1β) after exposure to WSE. The action mechanism could be the stimulation of the Nrf2 transcription factor by the bioactive compounds present in septum, followed by the downregulation of the NF-κB pathway and inhibition of pro-inflammatory cytokine secretion [12].

The anti-inflammatory effect of septum extract was also evaluated in an antitussive animal model using 3-month-old, healthy male Wistar rats [36]. The WSE treatment statistically decreased two tested inflammatory biomarkers, IL-6 and the receptor CXC-R1 for IL-8, in the lung tissue homogenates, and had no effect on the other receptor for IL-8 or CXC-R2. Another anti-inflammatory biomarker assessed in the lung tissue homogenates, NO concentration, revealed no significant changes after WSE treatment. The histopathological analysis of lung tissues at the level of alveolar parenchyma disclosed the largest area occupied by the alveolar spaces in the WSE group compared to both the control and codeine-treated groups in this citric acid aerosol-induced cough experimental model in rats [36]. The bioactive phytochemical content in septum manifesting antioxidant and anti-inflammatory activities could be part of the mechanism for the positive effects.

Several other extracts of *J. regia* matrices, including leaves, bark, husk, and kernels, exhibited inhibition of inflammatory processes and downregulation of the production of inflammatory mediators including leukotrienes [50]. Walnut oil reduced LPS-induced neuroinflammation, a condition related to cognitive decline and neurodegenerative diseases, in rat microglial cells by lowering nitrite generation, as well as COX2 and iNOS expression, in a concentration- and time-dependent manner [51]

### 4.3. Antidiabetic Activities

The diabetic pathology relates to hyperglycemia that causes health problems associated with ROS generation, glycation of proteins, lipoperoxidation, or insulin sensitivity, resistance, and secretion. Type 2 diabetes mellitus (T2DM), a chronic metabolic condition more common in the adult overweight and obese population, can lead to CVD, diabetic retinopathy, kidney failure, or mental health disorders [52]. An effective preventive and therapeutic strategy against diabetes is the improvement of blood glucose homeostasis through several interventions [53,54].

One of the measures is the inhibition of the activity of α-amylase and α-glucosidase, two typical postprandial digestive enzymes. Natural bioactive compounds that could inhibit these enzymes have recently been investigated because acarbose, an effective inhibitor, can cause side effects such as gastrointestinal symptoms. The action mechanism was related to the monosaccharide composition, polysaccharide molecular weight, spatial structure, or glycosidic bonds in a microwave-assisted polysaccharide fraction extracted from the septum that inhibited α-amylase and α-glucosidase. Furthermore, the same septum extract significantly lowered blood glucose levels in male ICR mice with streptozotocin-induced diabetes [29].

The α-glucosidase inhibitory activity of WSE was also investigated by Tan et al. [38]. The main bioactive compounds presenting inhibitory effects were four flavonoids, taxifolin, (+)-catechin, quercetin, and luteolin, with IC_50_ values in the range of 29.47–54.82 μM, stronger than the positive control acarbose (60.01 μM). Thus, the flavonoids through the structures of the A, B and C rings might be the primary molecules contributing to the α-glucosidase inhibitory activity of WSE. This evidence is in agreement with another study in which catechin and three quercetin glycosides, namely quercitrin (quercetin-3-O-rhamnoside), isoquercitrin (quercetin 3-β-D-glucoside), and hyperoside (quercetin 3-D-galactoside), the bioactive molecules isolated from walnut septum, could obstruct the activity of α-glucosidase [12].

Further phytochemical analysis isolated four new taraxasterane-type triterpenes in walnut septum, compounds that could suppress α-glucosidase [39].

The antihyperglycemic activity of septum was studied in diabetic mice induced with streptozotocin. After 4 weeks, the aqueous septum extract significantly reduced blood glucose (*p* < 0.05), but did not affect pancreatic structure [14]. Besides decreasing blood glucose levels, the aqueous septum extract inhibited hepatic damage in streptozotocin-induced diabetic mice [55].

In streptozotocin-induced diabetic rats, ethanolic septum extracts also decreased their blood glucose levels, without changing the insulin levels in diabetic and non-diabetic rats, and nor the normoglycemia in healthy, non-diabetic rats [16]. The hypoglycemic effect of septum was also analyzed in alloxan-induced diabetic rats. After 28 days, the ethanol septum extract significantly decreased blood glucose (*p* < 0.001), triglyceride levels (*p* < 0.05), and attenuated the low-density lipoprotein and total cholesterol [56].

The hypoglycemic capacity of WSE was evaluated in young rats, treated with D-gal to induce aging, and in old rats. Glycemia was significantly decreased in young rats compared to the D-gal group (*p* < 0.01) and in old rats compared to the control (*p* < 0.05) [31].

In a recent in vivo study, Zhang et al. combined a high-fat diet with low-dose streptozotocin and assessed the preventive effects of septum extract and regulation of gut microbiota on diabetic rats [57]. For the first time, it was indicated that supplementing with septum extract in advance could halt the onset of T2DM and inhibit metabolic disorders by reducing insulin resistance, liver damage, aberrant lipid metabolism, oxidative stress, and inflammation. The main components found in septum that also modulated the intestinal flora in rats were phenolic acids (rugosin F isomer, gallic acid, phlorizin, p-coumaric acid, vanillic acid, quercetin), flavonoids, and quinones [57].

Besides *J. regia* septum, the effects on blood glucose of other matrices from this species were also evaluated. Boulfia et al. showed that macerated acetone and ethanol extracts from walnut bark inhibited the activities of alpha-amylase and alpha-glucosidase [58]. These effects may be due to the various polyphenolics that were reported in walnut husk, buds, and bark [59]. The hypoglycemic activity of these bioactive compounds in terms of absorption, which could be influenced by the chemical and enzymatic changes during digestion and the plant matrix, was shown to be stronger for walnut green husk compared to brown shell [60].

Preclinical and clinical studies assessed the influence of walnut leaves. Thus, in streptozotocin-induced diabetic rats, an extract of walnut leaves decreased fasting blood glucose and MDA levels, improved the lipid profile, and significantly increased antioxidant enzyme activity in a dose-dependent manner compared to control [61]. The results of a randomized, double-blind, clinical trial of diabetic patients receiving hydroalcoholic walnut leaf extracts disclosed that glycemia and insulin resistance conditions were not modified, but other parameters, such as body mass index and systolic blood pressure, significantly decreased compared to baseline [62].

### 4.4. Antimicrobial Properties

Many bioactive molecules are known for their immunostimulant and antimicrobial activities [63]. Recent reports ascribed the antimicrobial inhibitory potential of *J. regia* to its rich and diverse phytochemical content [64]. Vieira et al. reported that the hydroethanolic extract of walnut green husk displayed good antimicrobial activities against Gram-positive and some Gram-negative bacteria, with the exception of *Pseudomonas aeruginosa* [65]. However, another study revealed that walnut extract, in a dose-dependent manner, could present activity against biofilm-related infections caused by this microorganism [66]. The macerated acetone extract from walnut bark presented antimicrobial capacity against *Staphylococcus aureus*, *Bacillus subtilis*, *Proteus mirabilis*, *P. aeruginosa*, *Escherichia coli*, and *Listeria innocua* [58], while the walnut pellicle extract displayed antibacterial and anti-biofilm activities in a dose-dependent manner [67].

The antimicrobial properties of the walnut extracts could be related to the polar part compounds, such as phenolic acids, and to the non-polar part compounds including tocopherols, sterols, monoterpenes, and juglone [68]. The inhibition of microbial activity was also related to vegetative phases [69].

Several experiments exposed that microbial strains have different sensibilities to septum extract. For example, walnut septum hydroalcohol extract incorporated in traditional butter presented good antimicrobial activity and inhibited the growth of the microorganisms (Coliforms, Psychrotrophic bacteria, yeasts, and molds) used in the study, excepting *S. aureus* [28].

The results of Genovese et al. showed that Gram-positive compared to Gram-negative bacteria were more sensitive to the WSE action, perhaps due to the wall structure and composition of bacteria [30]. Similar to these findings, another study showed that WSE presented strong antibacterial activity against *S. aureus* and a low effect against Gram-negative *E. coli* [12]. However, the extract was active against the other Gram-negative bacteria (*P. aeruginosa* and *Salmonella enteritidis*) and had no effect on the two fungi (*Candida albicans* and *Candida parapsilosis*). The results of a new study confirmed that WSE inhibited Gram-positive bacteria (*S. aureus* and *B. subtilis*) and again did not affect *E. coli* [70].

Besides phenolic compounds, Meng et al. highlighted that polysaccharides from septum, in a concentration-dependent manner, could also present antibacterial activity by disrupting the permeability of the cell wall and membrane of bacteria and inhibiting their growth [21]. Other studies also proved that the types of the extraction methods could influence the phytochemical profile of septum extracts and their biological effects [33].

### 4.5. Enzyme Inhibitory Activities

Tyrosinase is a copper-containing enzyme that controls the production of melanin. Thus, decreasing tyrosinase activity can prevent conditions related to the hyperpigmentation of the skin including melasma and age spots [71]. A hydroacetone septum extract revealed good tyrosinase inhibitory activity [24]. Previous studies reported that kernel and bark walnut extracts exhibited strong tyrosinase inhibition effects [50,72].

Among the therapies against obesity, a leading health issue linked to many cardiometabolic diseases is the inhibition of pancreatic lipase, an important enzyme involved in the hydrolysis of dietary fats. The inhibition of this enzyme may reduce overweight and obesity and regulate hyperglycemia [73]. One study analyzed the inhibitory capacity of septum extract [12]. It revealed an inhibition capacity of 50.79%, which was lower than those of walnut, pistachio, and pecan at 57, 68, and 69%, respectively [74].

The cholinergic system was proposed to be the most affected in neurodegenerative diseases, such as Alzheimer’s and Parkinson’s diseases. Therefore, one potential therapeutic strategy in these conditions is to increase the brain cholinergic levels by inhibiting the activities of AChE and butylcholinesterase (BuChE) [75].

Although some plant methanolic extracts including *Citrus limon*, *Ocimum basilicum*, and *Mentha spicata* displayed concentration-dependent inhibition of AChE and BuChE [76], WSE revealed no AChE inhibitory potential [12]. However, a recent study demonstrated that treatment with walnut hull extract could preserve antioxidant activity and significantly lower the cortical architectural damage in isoprenaline-induced pathological damage in rat brain tissue [77].

### 4.6. Antitumor Properties

In the past few decades, as the average life expectancy increased substantially, the prevalence of many chronic age-related diseases, including cancer, also increased [78]. Scientific evidence demonstrated that engaging in a healthier lifestyle or employing innovative curative treatments were key preventive and therapeutic strategies against these disorders [79]. Several studies revealed the beneficial effects of nut intake in lowering all-cause mortality and decreasing cancer incidence and mortality [80,81]. Walnut consumption in particular reduced the growth and survival of cancer cells via modifying the expression of cancer genes in mice and humans [82]. In addition, green walnut husk extracts inhibited the survival, migration, and invasion of gastric cancer cells in vivo and induced apoptosis, while in vitro they reduced cancer cell growth [57]. Juglone and its derivatives, found in most walnut matrices, play significant roles in cancer cell apoptosis, induction of autophagy, inhibition of cancer cell migration and proliferation [83].

Polysaccharides in the septum extract also showed antitumor activity, suppressing the proliferation of the human hepatocellular carcinoma cell line (HepG2) and gastric carcinoma cell line (BGC-823) in a dose-dependent manner (8–500 μg/mL) [23]. In a recent study, the antiproliferative activity of septum aqueous decoction was manifested not only against cancerous cells but also against normal mesenchymal stem cells, thus, further investigations are warranted [33].

Meanwhile, Genovese et al., investigating the cytostatic and cytotoxic effects of walnut septum extract on the human A172 glioblastoma cell line, showed that only the highest dose of the extract (140 μg/mL at 24 and 48 h) reduced the viability of non-cancerous human foreskin fibroblast (HFF-1) cells [30]. Yet, a reduction in the viability of cancerous cells was noted in a dose-dependent manner, from 35 to 140 μg/mL, with a significant reduction being observed at a WSE concentration of 70 μg/mL. Further cytostatic activity evaluation demonstrated a reduced ability of treated cancerous cells to proliferate and migrate. Moreover, after 48 h of the WSE treatment, caspase-3 activity showed a significant increase, which resulted in cancerous cell apoptosis. The septum phytochemical analysis disclosed several bioactive compounds that could be responsible for the antitumor activity including p-coumaric acid hexoside, quercetin 3-O-glucoside, and quercetin 3-O-rhamnoside (quercitrin), which are all predicted to be pro-apoptotic agents through caspase-3 expression stimulation, as well as epigallocathechin and epigallocathechin gallate, two flavonoids expected to promote apoptosis [30]. In addition to these molecules, other bioactive compounds found in plants were associated with a lower risk of cancer [84].

A recent in vitro experiment analyzed the WSE cytotoxicity on three cancerous cell lines (human lung adenocarcinoma cell A549 and breast cancer cells T47D-KBluc and MCF-7) and one normal cell line (human gingival fibroblasts HGF) [12]. Significant cytotoxic activity was observed in A549 and T47D-KBluc after 48 h of WSE treatment in a dose-dependent manner at concentrations greater than 75 g/mL. However, the MCF-7 line was not affected by the extract, while the normal cell line was more resilient to the WSE cytotoxic activity. The flow cytometry assay confirmed that the WSE treatment induced dose-dependent cytotoxicity in the evaluated cell lines, with the process involved proving to be mainly necrosis while apoptosis had a minor contribution [12].

Some of the anti-cancer mechanisms initiated by bioactive molecules include the regulation of signaling pathways in cancer cells, such as p53, NF-κB, MAPK, and PI3K/AKT, as well as the control of the activity of oncogenic and tumor suppressor ncRNAs [85,86]. Many dietary polyphenols present in septum extracts can positively modulate some of these mechanisms. Moreover, some of them, including rutin, quercetin, myricetin, and ellagic acid, can inhibit glucose uptake in some cancerous cells, such as Caco-2 cells, thus, being able to arrest cancer advancement [87].

### 4.7. Antitussive Effect

In a recent study, the antitussive potential of an optimized WSE was investigated in a citric acid-induced cough model by exposing Wistar rats to citric acid aerosols (17.5%) [36]. In the WSE treated-animals at a dose of 134 mg GAE/kg bw/day, the number of coughs decreased by approximately 68% versus negative control, a comparable effect to codeine. In addition, the WSE treatment significantly increased the latency of coughs and lowered pulmonary inflammation by reducing ROS and oxidative stress. Similarly, other plant extracts revealed antitussive, expectorant, and anti-inflammatory properties in rodent models [88]. Higher antitussive or expectorant effects were observed for *Thymus* spp., *Tilia* spp., *Nigella sativa*, *Linum usitatissimum*, or *Artemisia absinthium* [89]. As mentioned before, phytochemicals from walnut septum could act as antioxidants and antitussive agents. Hence, quercetin and its glycosides were reported to possess antioxidant potential and health benefits in age-related diseases [90,91]. Naringenin and its suspensions improved the antitussive and expectorant outcomes, and decreased cough frequency [92]. Other phenolic acids and flavonoids could also be potential treatments against asthma and chronic bronchitis [93].

### 4.8. Myelopoiesis Activation

Several medical procedures, such as chemotherapy and radiotherapy, are followed by negative side effects including leukopenia and reduced immune system function. Likewise, cyclophosphamide is linked to leukopenia, cytotoxicity, and immunosuppressive effects in a dose-dependent manner. In mice, after receiving a single injection of cyclophosphamide, the number of leukocytes significantly decreased. However, the treatment with aqueous septum extract stimulated the division, differentiation, and maturation of myeloblasts and lymphoblasts, thus showing an immune correction capacity through myelopoiesis activation [17]. Moreover, sorption capacity of erythrocytes increased in the blood of experimental mice that received septum and cyclophosphamide treatment compared to mice on cyclophosphamide only. The mechanism could be that the glutathione-dependent system, one of the key erythrocyte antioxidant systems, known for binding and detoxifying cyclophosphamide metabolites, might be activated by the bioactive compounds in septum. Experimental results also exposed that septum extract had a preventive outcome on erythrocyte hemolysis [13].

### 4.9. Anti-Aging Potential

Aging, oxidative stress, and chronic inflammation are considered major risk factors for most chronic disorders and age-related diseases. Oxidative stress and inflammation significantly impact aging, obesity, diabetes, cardiovascular and neurodegenerative diseases, and cancer [94]. Recent findings emphasized the potential of nutrition, i.e., the intake of bioactive antioxidant compounds, in preventing these conditions and extending health into aging [95]. For example, the consumption of walnuts was shown to decrease the levels of ROS and inflammatory cytokines, modulate the Nrf2/EpRE, PI3K/Akt/mTOR, and NF-κB signaling pathways, prevent mitochondrial dysfunction, inhibit carcinogenesis, and regulate energy homeostasis [81]. Thus, walnut intake could improve and extend the health span and lifespan. Many bioactive compounds in walnuts are responsible for these beneficial effects. Moreover, in vivo studies on animal aging models that were mentioned (Table 6) and discussed above demonstrated the anti-aging time-dependent exposure potential of WSE administered repeatedly [31]. With the knowledge that the phytochemical profiles of different *J. regia* matrices are almost identical, we can extrapolate that the biological activities noticed in this review after septum extract treatment, identified walnut septum as a potential candidate of healthy aging. 

## 5. Conclusions

This systematic review surveyed the published studies on walnut septum, one of the *Juglans regia* by-products, known and used in folk medicine for its medicinal properties.

Recent investigations described the phytochemical composition of this plant matrix and revealed a diverse and rich profile including polyphenols, among them phenolic acids, flavonoids, tannins, lignans, as well as carbohydrate and lipidic constituents. Other identified molecules were glutamate, histidine, and lysine, from amino acids, and potassium, calcium, and magnesium amid the minerals. Various in vitro and in vivo experiments highlighted the antioxidant and anti-inflammatory effects of walnut septum extracts. Besides, some bioactive compounds were shown to control α-amylase and α-glucosidase, postprandial digestive enzymes, or lower glycemia in rodent model studies, demonstrating a significant preventive role in diabetes. Furthermore, p-coumaric acid hexoside, quercetin 3-O-glucoside, quercetin 3-O-rhamnoside, epigallocathechin, and epigallocathechin gallate promoted apoptosis via activating caspase-3 expression, while quercetin, rutin, myricetin, and ellagic acid inhibited glucose uptake in cancerous cells and slowed cancer progression. Other experimental outcomes exposed the antimicrobial effects, AChE and tyrosinase inhibition properties, antitussive potential, or immune correction capacity through myelopoiesis activation. Thus, walnut septum, through lowering oxidative stress, inhibiting inflammation, and modulating several important signaling pathways, could have anti-aging potential and play a role in healthy aging, a key attribute of lifespan.

We consider walnut septum to be an important biological matrix, a rich natural source of bioactive compounds that deserves to be investigated in the future in order to be fully exploited in the food, cosmetic, or pharmaceutical industry.

## Figures and Tables

**Figure 1 antioxidants-12-00604-f001:**
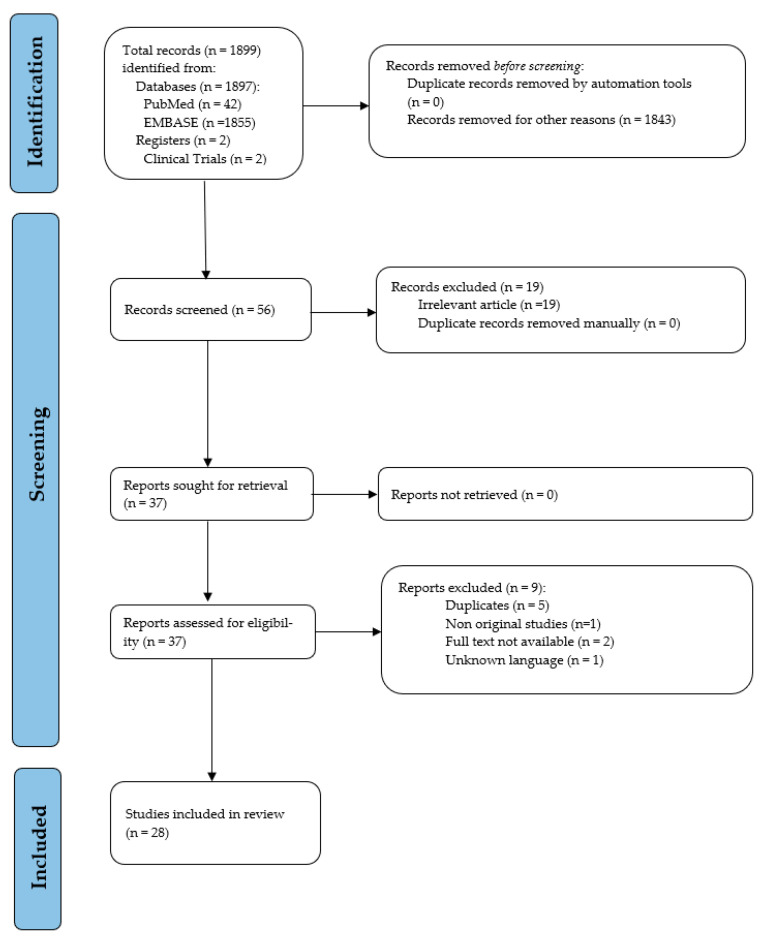
PRISMA flow diagram of study selection.

**Figure 2 antioxidants-12-00604-f002:**
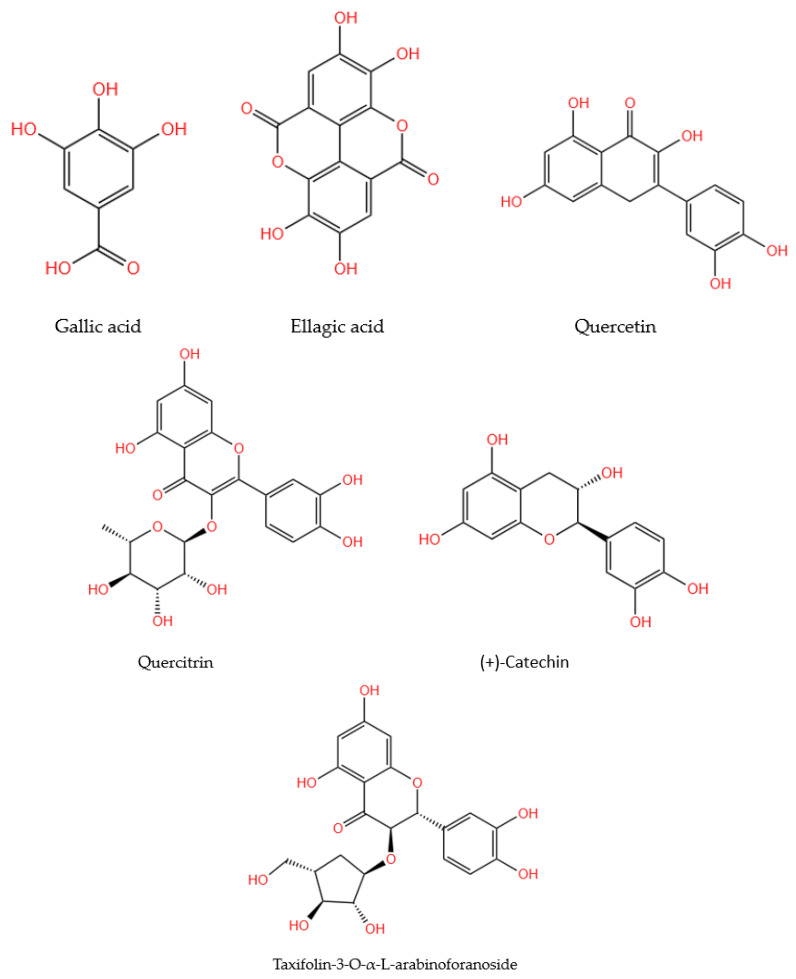
Chemical structures of the main phenolic compounds identified and quantified in walnut septum.

**Figure 3 antioxidants-12-00604-f003:**
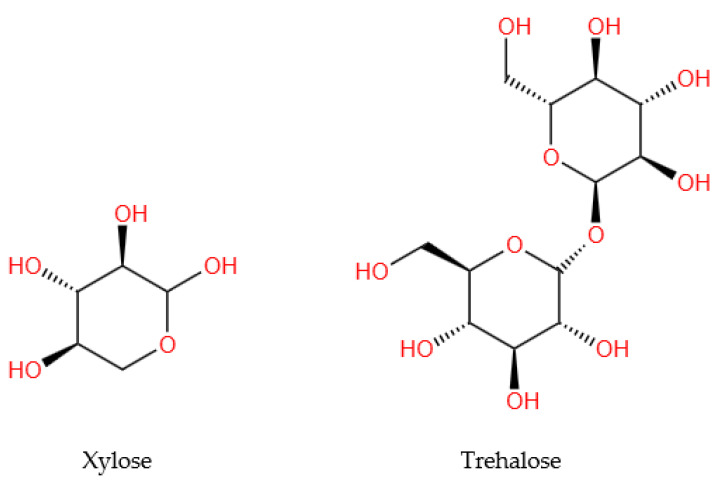
Chemical structures of the main carbohydrate compounds quantified in walnut septum.

**Table 1 antioxidants-12-00604-t001:** Characteristics of the selected studies.

Ref.	Country	Study Type	Study Purpose	Materials/Type of Extracts	Analysis Metods/Biological Systems/Animal Models	Study Outcomes/Biological Activities
[16]	Iran	In vivo	Hypoglicemiant effect of WSE	WSE—90% ethanol	Streptozotocin-induced diabetic rats (60 mg/kg bw)	WSE (gavage: 100, 200, 500, 1000 mg/kg bw, 2 weeks) significantly decreased the blood glucose level.
[14]	Iran	In vivo	Hypoglicemiant effect of WSE; Histopathological pancreatic structure	WSE—aqueous	Streptozotocin-induced diabetic mice (220 mg/kg bw)	WSE (oral; 200, 400, 600, 800 mg/kg bw, 4 weeks) reduced blood glucose level. No effect on pancreatic structure.
[17]	Georgia	In vivo	Immune corrective properties of WSE on leukopenia induced experimental model	WSE—aqueous (2:1, water:WS)	Cyclophosphamide—induced leukopenia mice (350 mg/kg bw)	WSE stimulates myelopoiesis in mice with leukopenia induced by chemotherapy.
[18]	Iran	In vivo	The antidiabetic and hypolipidemic effects of WSE on diabetic rats	WSE—1:10 *w*/*v* in 80% ethanol	Alloxan-induced diabetic rats (150 mg/kg bw)	WSE (oral: 200, 400 mg/kg bw, 4 weeks) significantly decreased blood glucose, ALAT, ASAT, TG; WSE (200 mg/kg bw) attenuated TC and LDL-C.
[13]	Georgia	In vivo	The effect of WSE on the functional characteristics of erythrocytes during the administration of cyclophosphamide	WSE—2:1 (water:WS)	Cyclophosphamide—cytotoxicity induced mice (350 mg/kg bw)	WSE had a preventive effect on the process of hemolysis; WSE increased sorption capacity and resistance of erythrocytes to lysis.
[19]	Iran	In vivo	Acute and subacute toxicity of WSE; Effect of WSE on oxidative stress and enzymes; Histopathology	WSE—95% methanol	Female Wistar rats treated with WSE (oral: 10, 100, 1000, 1600, 2900, and 5000 mg/kg bw and gavage: 1000 mg/kg bw, 4 weeks)	WSE showed no acute or subacute adverse effects with dose of 1000 mg/kg bw (oral by gavage). WSE may improve kidney structure and function.
[20]	China	In vitro	Isolation of taxifolin-3-O-arabinofuranoside isomers from WSE	WSE—70% ethanol	Compounds identification: silica gel column chromatography (CC), Sephadex LH-20 CC, ODS CC, and preparative HPLC	(2S,3S)-taxifolin-3-O-α-d-arabinofuranoside and its isomer (2S,3S)-taxifolin-3-O-α-l-arabinofuranoside were isolated from WSE.
[21]	China	In vitro	Antioxidant and antibacterial activities of polysaccharides from WS (WSP)	WSP—water-soluble, purified	Composition: HPIC; Antioxidant activity: DPPH, ABTS, -OH, FRAP. Antibacterial activity: on *E. coli, P. aeruginosa, S. aureus, E. faecalis* strains	WSP mainly contained glucose, galactose, arabinose, xylose, and traces of mannose. WSP showed antioxidant and antibacterial activities in vitro.
[22]	China	In vitro	Chemical constituents and anti-inflammatory effects of WSE	WSE—ethyl acetate extract;	Isolation: pre-HPLC, HSCCC; identification: NRM, ESI-MS; Lipopolysaccharide (LPS)-stimulated RAW 264.7 macrophages model (in vitro)	Fourteen compounds isolated; Dihydrophaseic acid, blumenol B and (4S)-4-hydroxy-1-tetralone for the first time in WS; Gallic acid, ethyl gallate. and (+)-dehydrovomifoliol—anti-inflammatory activity.
[23]	China	In vitro	Antitumor, and immune-enhancement effects of polysaccharides from WS (WSP)	WSP—water-soluble, purified (Water/raw material ratio: 20 mL/g)	Cytotoxicity (MTT assay) on Murine macrophage cell line RAW 264.7, HepG2, and BGC-823 cells. Immune-enhancement activity: cytokines production, RT and RT-PCR on RAW 264.7 cells	WSP could effectively suppress the proliferation of tumor cells. WSP could significantly enhance phagocytosis, stimulate the production of NO, TNF-α, and interleukins (IL-6 and IL-1β), and promote the mRNA expression levels of iNOS, TNF-α, IL-6, and IL-1β in a dose-dependent manner.
[24]	Romania	In vitro	Process optimization for improved phenolic compounds recovery from WS Phytochemical profile and biological activities	WSE—70% ethanol	Phytochemical profile: UTE, LC-MS/MS; Quantitative det.: TPC, TFC, CTC. Antioxidant activity: ABTS, DPPH, FRAP assays; Tyrosinase inhibitory effect	The content in phenolic compounds, tannins, and phytosterols was correlated with the evaluated antioxidant and tyrosinase inhibitory activities.
[25]	China	In vitro	Phenolic composition and nutritional attributes of WS	WS—dried under 35 °C in oven, grinded	Identification: minerals (ICP-MS), fatty acids (GC/MS), amino acids, monosaccharides, phenolic compounds (HPLC); Phenolic compounds extract: HPLC–DAD/ESI–MS	Phenolic acids, hydroxybenzoicacid, isoflavone, and flavone were identified in the free phenolic fractions (FPFs) of WS.
[26]	China	In vitro	Characterization of phenolics in WSE	WSE—70% methanol	Analysis of phenolic compounds: HPLC-UV; Individual phenolic compounds: UPLC-MS; TPC—Folin–Ciocalteau method	A total of 75 phenolic compounds were identified: flavonols, flavanols, procyanidins, ellagitannins, gallotannins, phenolic acids and related compounds
[27]	China	In vitro	Identification and quantification of bioactive compounds in WSE	WSE—acetone/cyclohexane (2:1, *v*/*v*)	Identification: hydrolyzable tannins, flavonoids, phenolic acids, quinones: UHPLC-Q-Orbitrap HRMS; Polyphenols quantification: UHPLC-MS/MS	Two-hundred compounds, including hydrolyzable tannins, flavonoids, phenolic acids, and quinones, were identified. Total contents of polyphenols: 2.88−6.18 mg/g.
[28]	Iran	In vitro	Antioxidant and antimicrobial effect of (WSE) in traditional butter	WSE—hydroalcoholic extract	Antioxidant effect: DPPH, reducing power, total phenol assays; Physicochemical analysis; Microbiological analysis; Sensory evaluation	The utilization of WSE is safe in the 0.05%, 0.1% and 0.5% conc. and has benefits for oxidative stability and inhibited the microbial growth of butter stored under suitable conditions.
[29]	China	In vitro; In vivo	The hemolysis inhibitory effects of DJP-2; Antioxidant effect of WSP (in vitro); Hypoglycemic activity (in vivo); Antiglycation activities.	WSP- water-soluble, purified	H_2_O_2_-induced hemolysis of RBCs from Kunming mice; Hepatic L02 cells model; Type 2 diabetes mellitus mice model (male ICR mice, ip. STZ 1%) Hypoglycemic assay (α-amylase and α-D-glucosidase activity)	WSP showed remarkable hemolysis inhibitory activity. WSP markedly weakened the oxidative damage induced by H_2_O_2_ in hepatic L02 cells via strengthening the cell viability. WSP showed hypoglycemic activities in vivo and in vitro. WSP (3 mg/mL) exerted more significant antiglycation activities than aminoguanidine.
[30]	Italy	In vitro	The cytostatic potential and antibacterial activity of WSE	WSE—96% ethanol	Human glioblastoma cells (A172). Clinical isolates: Gram-positive and Gram-negative strains	WSE (70 μg/mL) caused a significant reduction of healthy A172 cells and an increment of apoptotic and necrotic cells. WSE (275 μg/mL) affected the growth of *G-pos* and *G-neg* bacterial strains, resistant to Ciprofloxacin.
[31]	Romania	In vivo	Antioxidant effect of WSE in aged induced and old rats	WSE—water: acetone (50:50, *v*/*v*)	D-Galactose-induced aging model (Wistar female rats, D-gal 1200 mg/week), old rats; Cellular antioxidant status: ABTS, DPPH, FRAP; Oxidative stress biomarkers: OS, AGEs, NO, MDA, GSH; AChE activity; Histopathological and immunohistochemical analyses	A significant improvement in cellular antioxidant activity and decrease of ROS, advanced glycation end products, NO, MDA, or increase of glutathione after WSE intake.
[12]	Romania	In vitro	Assessment of bioactive molecules and in vitro biological effects of WS	WSE WS mixed with water/acetone (50:50, *v*/*v*) at a ratio of 1:10 (*w*/*v*)	Tocopherol content: LC-MS/MS; Biological activities: enzyme, cholinesterase, α-glucosidase, and lipase inhibitory activity; Antibacterial and antifungal activity: *G. pos*, *G. neg*. bacteria, and fungi strains; Antimutagenic assay: Salmonella typhimurium TA 98 and TA 100 strains; Cytocompatibility: cancerous A549 and T47D-KBluc cells; Antioxidant potential: ROS in A549, MCF-7 and HGF cell types; Anti-inflammatory potential in HGF cells	WSE has inhibitory effect on acetylcholinesterase, α-glucosidase, and lipase activity. WSE—strong antimicrobial potential against Staphylococcus aureus, Pseudomonas aeruginosa, and Salmonella enteritidis. Strong antimutagenic inhibitory effects against TA 98 and TA 100 strains. WSE—cytotoxic, antioxidant, and anti-inflammatory activity (IL-6, IL-8, IL-1β) in the selected cell lines.
[32]	China	In vitro	Comparison of phenolic compounds extracted from WS, walnut pellicle (WP), and flowers of *Juglans regia* (WFl)	ME—methanolic condensation reflux extraction UE—ultrasonic waves extraction EE—enzymes assisted-extraction	TPC—Folin-Ciocalteu method, FL, CT; Antioxidant activity: OH radical scavenging, DPPH, ABTS, Fe2+ chelating	A total of 50 phenolic compounds were identified by LC-MS/MS. The antioxidant capacity of WP was stronger than that of WFl and WS.
[33]	Iran	In vitro	The effect of the DISWK on the survival, TAC, and cell cycle of rat bone marrow. The expression of genes involved in pancreatic insulin-producing β-cell (IPCs) commitment and glucose uptake in MSCs-derived IPCs.	DISWK—aqueous	Survival, TAC, and cell cycle on rat bone marrow-derived mesenchymal stem cells (rBM-MSCs). The expression of genes involved in pancreatic insulin-producing β-cell (IPCs) commitment and glucose uptake in MSCs-derived IPCs	DISWK significantly reduced the viability of rBM-MSCs and decreased their TAC in long-term cultures. DISWK suppressed the rBM-MSCs cell cycle at S and G2 phases. DISWK significantly upregulated Ins1/2, Insr, and Glut1 genes and downregulated Pdx1 gene in the rBM-MSCs-derived IPCs.
[34]	China	In vivo	Separation and identification of antioxidant chemical components in WS and functional evaluation in *Caenorhabditis elegans*	WSE—ethanol and water	*C. elegans*—to evaluate antioxidant capacity and examined the changes in metabolome by GC-TOF/MS	Four monomers were isolated, three for the first time in WS CDG could improve oxidative stress and prolong lifespan in *C. elegans* by improving the activities of antioxidant enzymes and reducing ROS and MDA content.
[35]	China	In vitro; In vivo	Anti-fatigue and antioxidative effects of water and alcohol extracts from WS	WSE—water (1:20, *w*/*v*); WSE—70% ethanol	H_2_O_2_-induced oxidative stress in HepG2 cells; Adult male Kunming mice—behavioral tests and biochemical assay	Both extracts—HepG2 protective capacities; capable of scavenging DPPH and ABTS as VitC
[36]	Romania	In vivo	Antitussive, antioxidant, and anti-inflammatory effects of a WSE	WSE—water: acetone (50:50, *v*/*v*)	Citric acid aerosol-induced cough experimental model in rodents (Wistar male rats)	A significant antitussive effect of WSE, superior to codeine; The antioxidant and anti-inflammatory effects of WSE were confirmed by biochemical assays and histopathological analysis.
[37]	Greece	In vitro	Determination of flavonoids in by-products of plant origin: An application for the valorization of the WS	WS	A UAE-SPE-HPLC-DAD analytical method was developed and validated	Seven flavonoids were determined: catechin, rutin, myricetin, luteolin, quercetin, apigenin, and kaempferol.
[38]	China	In vitro	Chemical constituents of WS; The hypoglycemic effect of WSE	WSE—70% ethanol	Identification of constituents: LC–MS, HPLC Hypoglycemic assay—α-glucosidase activity (in vitro)	37 constituents identified; the most abundant: rugosin F isomer, gallic acid, phlorizin, p-coumaric acid, vanillic acid, and quercetin. Flavonoids: the main bioactive substances contributing to the α-glucosidase inhibitory activity of WSE.
[39]	China	In vitro	Phytochemical investigation on WSE	WSE—70% ethanol	Identification of constituents: HRESIMS, NMR and X-ray single-crystal diffraction analysis; Molecular docking: α-glycosidase inhibitory activities	The isolation of four new taraxasterane-type triterpenes. The structure-activity relationship of the bioactive taraxasterane-type triterpenes against the α-glucosidase molecular docking was established.
[40]	China	In vitro	The effect of WSE on the oxidative stability of soybean oil during deep frying.	WSE—80% ethanol	TPC—Folin-Ciocalteu method; Quantification of bioactive compounds: Q Exactive Orbitrap mass spectrometer coupled to a Vanquish ultraperformance liquid chromatography system; Determination of oxidized and polymeric TAGs: HPSEC method; Quantification of oxidation products: ^1^H NMR	31 polyphenols determined; the main components: catechin, quercitrin, taxifolin, quercetin 3-β-d-glucoside, epicatechin, gallic acid, and 3,4-dihydroxybenzoic acid. The antioxidants delayed the degradation of triglycerides and inhibited the increase in the contents of panisidine, oxidized triglyceride monomers, triglyceride dimers, and TG oligomers. WSE: better inhibitory effect on the formation of (E)-2-alkenals, (E,E)-2,4- alkadienals, 4-oxo-alkanals, primary alcohols, and secondary alcohols detected by ^1^H NMR TBHQ and TP

ABTS—2, 2′-azino-bis-3-ethylbenzothiazoline-6-sulfonic acid; AChE—Acetylcholinesterase; AGEs—advanced glycation end products; AKI—induced acute kidney injury; ALAT—alanine aminotransferase; ASAT—aspartate aminotransferase; CDG—2-carboxy-5,7-dihydroxy3-naphthyl-β-D-glucopyranoside; CTC—condensed tannin content; DJ—diaphragma juglandis; DISWK—decoction of the internal septum of the walnut kernel; DJP-2—diaphragma juglandis fructus polysaccharides; DPPH—2,2-diphenyl-1-picrylhydrazyl; ESI-MS—electrospray ionization mass spectrometry; FRAP—ferric reducing ability of plasma; GC/MS—gas chromatography–mass spectrometry; GC-TOF/MS—gas chromatography coupled to time-of-flight mass spectrometry; GSH—reduced glutathione; HPIC—high-performance ion chromatography; HPLC—high performance liquid chromatography; HPLC—high performance liquid chromatography; HPLC–DAD/ESI–MS—high performance liquid chromatography coupled with a photodiode-array detector and a mass spectrometry; HPLC-MS/MS—high-performance liquid chromatography-tandem mass spectrometry; HPLC-UV—high-performance liquid chromatography coupled with ultraviolet detector; HPSEC—high-performance size exclusion chromatography; HRMS—high-resolution mass spectrometry; HRESIMS—high resolution electrospray ionization mass spectroscopy; HSCCC—high-speed counter-current chromatography; HSCCC—high-speed counter-current chromatography; ICP-MS—inductively coupled plasma-mass spectrometer; IL-1β—interleukin-1β; IL-6—in-terleukin-6; iNOS—inducible nitric oxide synthase; IPCs—pancreatic insulin-producing β-cell; JRSME—Juglans regia septum of methanol extract; LC-MS/MS—liquid chromatography with tandem mass spectrometry; LDL-C—low density lipoprotein-cholesterol; MDA—malondialdehyde; mRNA—messenger ribonucleic acid; MTT—3-(4,5-dimethylthiazol-2-yl)-2,5-diphenyl-2H-tetrazolium bromide; NMR—nuclear magnetic resonance; NO—nitric oxide; ODS—overspeed detection system; OS—oxygen species; pre-HPLC—preparative high performance liquid chromatography; RBCs—red blood cells; RT—reverse transcription; RT-PCR—real-time reverse transcription–polymerase chain reaction; STZ—streptozotocin; SWS—septum of walnut shell = WS; SWSE—septum of walnut shell extract; TAC—total antioxidant capacity; TBHQ,—tert-butylhydroquinone; TC—total cholesterol; TFC—total flavonoid content; TG—triglycerides; TGPs—oxidized TAG polymers; TNF-α—tumor necrosis factor-alpha; TP—tea polyphenol; TPC—total phenolic content; UAE-SPE-HPLC-DAD—ultrasound-assisted protocol combined with solid phase extraction and coupled to high-performance liquid chromatography with diode array detection; UHPLC-Q-Orbitrap HRMS—ultrahigh-performance liquid chromatography coupled with hybrid quadrupole-Orbitrap high-resolution mass spectrometry; UPLC-MS—ultra-performance liquid chromatography coupled with mass spectrometry; UTE—ultra-turrax extraction; WHE—walnut kernel septum membranes hydroalcohol extract; WP—walnut pellicle; WS—walnut septum/septa; WSE—walnut septum extract; WFl—walnut flower.

**Table 2 antioxidants-12-00604-t002:** Phenolic compounds identified and quantified in walnut septum.

Ref.	Extract	Analytical Method	Compounds	Amount
[20]	70% ethanolic extract	LC-ESI-MS; HPLC-UV	*Flavonoids:* (2S,3S)-Taxifolin-3-O-α-d-arabinofuranoside, (2S,3S)-Taxifolin-3-O-α-l-arabinofuranoside	NQ
[22]	Ethanolic extract	LC-ESI-MS; HPLC-UV	*Phenolic acids*: Gallic acid, p-Hydroxybenzoic acid, Protocatechuic acid, Vanillic acid	NA
			*Flavonoids*: Quercitrin, Taxifolin (dihydroquercetin), Taxifolin-3-O-α-l-arabinofuranoside	NA
			Esters: Ethyl gallate	NA
[24]	Extract in water:acetone (50:50, *v*/*v*)	LC-ESI-MS; HPLC-UV	*Phenolic acids*: p-Coumaric acid, Ferulic acid, Gentisic acid	<LOQ
			*Flavonoids*: Hyperoside	67.32 µg/g dw
			Isoquercitrin	103.60 µg/g dw
			Quercitrin	1073.04 µg/g dw
			Quercetin	<LOQ
		LC-ESI-MS	*Phenolic acids*: Gallic acid	79.58 µg/g dw
			Protocatechuic acid	9.94 µg/g dw
			Syringic acid	5.20 µg/g dw
			Vanillic acid	5.57 µg/g dw
			*Flavonoids*: Catechin	597.65 µg/g dw
			Epicatechin	12.54 µg/g dw
[25]	Extracts in acetone and methanol	LC-ESI-MS; HPLC-UV	*Phenolic acids*: Gallic acid, Phtalic acid, Vanillic acid	NQ
			*Flavonoids*: Catechin, Kaempferol, Taxifolin, Taxifolin-3-O-α-l-arabinofuranoside, Quercitrin, Quercetin-3-O-(4-O-acetyl)-α-l-rhamnopyranoside,	NQ
			*Other polyphenols*: Vanillin	NQ
			*Esters*: Ethyl gallate, Propyl gallate	NQ
[27]	Extracts in acetone/cyclohexane (for identification) and 50% methanol (for quantification) obtained by UE	UHPLC-Q-Orbitrap HRMS	*Phenolic acids*: Ellagic acid	518.38–1733.64 μg/g dw ^a^
			Neochlorogenic acid	32.49–126.57 μg/g dw
			Gallic acid	89.87–219.09 μg/g dw
			Protocatechuic acid	44.28–154.04 μg/g dw
			Syringic acid	11.44–26.76 μg/g dw
			Vanillic acid	18.68–44.13 μg/g dw
			*Flavonoids*: (+)-Catechin	289.19–693.32 μg/g dw
			(−)-Epicatechin	8.82–36.44 μg/g dw
			(−)-Epigallocatechin gallate	0.43–2.21 μg/g dw
			(−)-Epicatechin gallate	63.47–194.79 μg/g dw
			Taxifolin	33.99–153.31 μg/g dw
			Myricitrin	11.22–28.13 μg/g dw
			Hyperoside	4.20–60.27 μg/g dw
			Isoquercitrin	20.89–116.68 μg/g dw
			Astilbin	9.19–31.62 μg/g dw
			Taxifolin-3-O-arabinofuranoside	519.60–2181.84 μg/g dw
			Quercitrin	145.42–983.58 μg/g dw
			Phlorizin	3.18–34.57 μg/g dw
			Quercetin	4.76–17.43 μg/g dw
			*Esters*: Methyl gallate	34.32–355.66 μg/g dw
			*Hydroxibenzaldehydes*: Protocatechualdehyde	9.79–34.60 μg/g dw
[26]	70% methanolic extracts obtained by UE	HPLC-UV	*Phenolic acids*: Chlorogenic acid, Coumaroylquinic acid isomers, Ellagic acid, Gallic acid, Neochlorogenic acid, p-Coumaroylquinic acid	NQ
			*Flavonoids*: (−)-Epicatechin, (+)-Catechin, Kaempferol-rhamnoside, Hyperoside, Quercetin-rhamnose-pentoside Quercetin-3-o-glucoside, Quercetin-rhamnoside, Quercetin-rhamnoside isomers, Quercetin-rhamnose-hexoside, Quercetin-pentoside isomers, Quercetin-galloylhexoside isomers B-type procyanidin dimer, trimer, tetramer, pentamer, hexamer, and isomers	NQ
			*Tannins*: Ellagic acid hexoside, Ellagic acid pentoside, Ellagic acid pentoside isomers, Digalloyl-glucose isomer, HHDP-glucose isomer, bis-HHDP-glucose, Galloyl-HHDP-glucose, Monogalloyl-glucose isomers, Digalloyl-glucose isomers, Trigalloyl-glucose isomers, Euprostin A isomer	NQ
[32]	70% methanolic extracts obtained by ME, UE, and EE	LC-MS/MS	*Phenolic acids*: Ellagic acid	239.65–351.92 µg/g dw ^b^
			Quinic acid	28.09–44.62 µg/g dw
			Coumaroylquinic acid	7.64–10.65 µg/g dw
			P-coumaric acid	19.82–28.74 µg/g dw
			Hydroxymandelic acid	3.58–5.01 μg/g dw
			Ferulic acid isomer	55.77–87.96 μg/g dw
			Protocatehuic acid	1.15–1.24 μg/g dw
			Caffeic acid	5.54–5.88 μg/g dw
			P-Hydroxybenzoic	7.06–8.54 μg/g dw
			*Flavonoids*: (+)-Catechin	2.29–3.65 μg/g dw
			Isoquercitrin	3.0–4.27 μg/g dw
			Luteolin	20.41–28.39 μg/g dw
			Myricitrin	21.34–31.05 μg/g dw
			Quercetin	38.01–39.84 μg/g dw
			Quercetin galloylhexoside isomer	8.30–13.80 μg/g dw
			Quercitrin	29.59–33.83 μg/g dw
			Taxifolin	11.75–15.68 μg/g dw
			Taxifolin-3-O-arabinofuranoside	7.17–13.84 μg/g dw
			*Tannins*: HHDP-glucose isomer	3.84–6.75 μg/g dw
			Galloyl-HHDP-glucose	4.21–5.77 μg/g dw
			Monogalloyl-glucose	3.20–4.87 μg/g dw
			Trigalloyl-glucose	2.64–4.90 μg/g dw
			*Esters*: Methyl gallate	3.54–10.60 μg/g dw
[34]	Ethanolic extract	GC-TOF/MS	*Phenolic acids*: p-Coumaric acid	NQ
			*Phenylpropanoids*: (1′-methyl-2′-hydroxy) propane-O-α-D-glucopyranoside, (4′–hydroxyphenyl) methylene-O-β-D-glucopyranosyl-(4 → 1)-α-L-arabinopyranoside, 2-carboxy-5,7-dihydroxy-3-naphthyl-β-D-glucopyranoside	NQ
[37]	Methanolic extracts (Chandler, Vina, and Franquette varieties) obtained by UE	HPLC-DAD	*Flavonoids*: Apigenin	1–10 µg/g dw ^c^
			Catechin	25–53 µg/g dw
			Kaempferol	2–9 µg/g dw
			Luteolin	1–5 µg/g dw ^c^
			Myricetin	1–11 µg/g ^d^
			Quercetin	4–16 µg/g dw
			Rutin	1–6 µg/g dw
[38]	70% ethanolic extracts	HPLC	*Flavonoids*: (+)-Catechin, Catechin lactone A, (2R)-eriodictyol-5-O-β-D-glucoside,3,5,7-trihydroxylchromone-3-O-α-L-arabinofuranoside, Luteolin, Naringenin 7-O-β-D—glucopyranoside, Quercetin, 3-O-methylquercetin, Avicularin, Quercetin-3-O-α-D-arabinofuranoside, Quercetin 3-O-β-D-xylopyranoside, Quercetin-3-O-(6″-O-galloyl)-β-D-galactopyranoside, Quercetin-3-O-β-D-glucopyranoside, Sakuranetin 5-O-β-D xylopyranoside, Taxifolin, Taxifolin-3-O-α-L-arabinofuranoside, Taxifolin-3-β-D-xylopyranoside	NQ
			*Phenylpropanoids*: 1,6-di-O-(E)-Coumaroyl-β-D-glucopyranoside,3,5,7-Trihydroxylchromone-3-O-α-L-arabinofuranoside, 1-O-(Z)-Coumaroyl,6-O-(E)-coumaroyl-β-D-glucopyranoside, Erythro-(7S,8R)-guaiacyl-glycerol-β-O-4′-dihydroconiferyl ether, 1-(4′-Hydroxy-3′-methoxyphenyl)-2-[4″-(3-hydroxypropyl)-2″,6″-dimethoxyphenoxy]propane-1,3-diol, Rosalaevin B	NQ
			*Lignans*: (5-methoxy-(+)-Isolariciresinol, Erythro-guaiacyl-glycerol-β-O-4′-(5′)-methoxylariciresinol, Rhoiptelol B, Dihydrodehydodiconiferyl alcohol	NQ
[40]	80% ethanolic extracts obtained by UE	UPLC-Q-MS	*Phenolic acids*: Gallic acid	272.52 ± 0.52 μg/g dw ^d^
			3,4-Dihydroxybenzoic acid	259.06 ± 0.09 μg/g dw
			Vanillic acid	66.06 ± 1.30 μg/g dw
			Syringic acid	37.94 ± 0.22 μg/g dw
			Benzoic acid	18.55 ± 0.07 μg/g dw
			4-Hydroxybenzoic acid	17.14 ± 0.10 μg/g dw
			Caffeic acid	9.81 ± 0.07 μg/g dw
			Salicylic acid	6.50 ± 0.28 μg/g dw
			Trans-ferulic acid	4.56 ± 0.19 μg/g dw
			p-Hydroxycinnamic acid	10.17 ± 0.58 μg/g dw
			Trans-cinnamic acid	0.74 ± 0.11 μg/g dw
			4-Hydroxy-3,5-dimethoxycinnamic acid	0.27 ± 0.06 μg/g dw
			Flavonoids: Catechin	9989.16 ± 0.73 μg/g dw
			Quercitrin	6816.18 ± 1.61 μg/g dw
			Taxifolin	569.39 ± 0.88 μg/g dw
			Quercetin 3-β-D-glucoside	399.00 ± 1.07 μg/g dw
			Epicatechin	362.10 ± 0.50 μg/g dw
			Quercetin	54.10 ± 0.06 μg/g dw
			Naringenin	31.84 ± 0.27 μg/g dw
			Rutin	17.32 ± 0.44 μg/g dw
			Vanillin	15.53 ± 0.45 μg/g dw
			Dihydrokaempferol	14.57 ± 0.17 μg/g dw
			Luteolin	7.60 ± 0.45 μg/g dw
			Dihydromyricetin	5.75 ± 0.12 μg/g dw
			Naringenin chalcone	3.34 ± 0.10 μg/g dw
			Vitexin	1.90 ± 0.19 μg/g dw
			Isorhamnetin	0.99 ± 0.14 μg/g dw
			Kaempferol	0.96 ± 0.03 μg/g dw
			Apigenin	0.36 ± 0.05 μg/g dw
			*Benzaldehydes*: Protocatechualdehyde	85.46 ± 0.63 μg/g dw
			Syringaldehyde	4.90 ± 0.03 μg/g dw

GC-TOF/MS—gas chromatography coupled to time-of-flight mass spectrometry; HPLC—high performance liquid chromatography; HPLC–DAD/ESI–MS—high performance liquid chromatography coupled with a photodiode-array detector and a mass spectrometry; HPLC-UV—high-performance liquid chromatography coupled with ultraviolet detector; LC-MS/MS—liquid chromatography with tandem mass spectrometry; LOQ—limit of quantification (0.5 µg/mL); NA—not available (the amount of bioactive compound is not expressed in relation to the mass of dry walnut septum); NQ—non-quantified; LC-ESI-MS—liquid chromatography coupled to mass spectrometry with electrospray ionization; UPLC-Q-MS—ultra-performance liquid chromatography coupled with Orbitrap mass spectrometry; UE—ultrasonic wave extraction; UTE—ultra-turrax extraction; WSE—walnut septum extract. ^a^—The concentration range for the ten walnut septum batches; ^b^—The concentration range for the three types of walnut septum extracts (EE, UE, ME); ^c^—The concentration range for the three walnut varieties (Chandler, Vina, and Franquette); ^d^—The amounts are presented as means ± SD (standard deviation); dw—dry weight.

**Table 3 antioxidants-12-00604-t003:** Lipidic compounds identified and quantified in walnut septum.

Ref.	Extracts	Analytical Method	Compounds	Amount
[24]	Acetone and ethanol extracts obtained by UTE	LC-MS	Phytosterols:	
			β-Sitosterol	31,018.16 µg/g dw
			Stigmasterol	<LOQ
			Campesterol	292.07 µg/g dw
[25]	Methanolic extract	GC/MS	*Fatty acids*:	
			Octanoic acid	1.37 ± 0.05 mg/kg dw *
			Decylic acid	1.86 ± 0.02 mg/kg dw
			Lauric acid	5.33 ± 0.10 mg/kg dw
			Myristic acid	11.62 ± 0.20 mg/kg dw
			Pentadecanoic acid	22.24 ± 0.26 mg/kg dw
			Palmitic acid	700.55 ± 8.80 mg/kg dw
			Margaric acid	28.95 ± 0.16 mg/kg dw
			Stearic acid	143.96 ± 4.02 mg/kg dw
			Arachidic acid	18.98 ± 0.27 mg/kg dw
			Heneicosanoic acid	18.66 ± 0.06 mg/kg dw
			Behenic acid	50.79 ± 0.31 mg/kg dw
			Tricosanoic acid	44.91 ±0.28 mg/kg dw
			Lignoceric acid	49.89 ± 0.24 mg/kg dw
			Palmitoleic acid	17.45 ± 0.24 mg/kg dw
			cis-10-Heptadecenoic acid	7.51 ± 0.14 mg/kg dw
			Oleic acid	549.92 ± 18.98 mg/kg dw
			Linoleic acid	1314.06 ± 10.71 mg/kg dw
			c-Linolenic acid	5.39 ± 0.09 mg/kg dw
			cis-11-Eicosenoic acid	5.47 ± 0.16 mg/kg dw
			cis-11,14-Eicosadienoic acid	3.90 ± 0.15 mg/kg dw
			cis-8,11,14-Poxyeicosatrienoic acid	0.77 ± 0.05 mg/kg dw
			Arachidonic acid	1.06 ± 0.05 mg/kg dw
			cis-5,8,11,14,17-Timnodonic acid	2.20 ± 0.08 mg/kg dw
			Erucic acid	1.82 ± 0.06 mg/kg dw
			cis-13,16-Docosadienoic acid	1.36 ± 0.09 mg/kg dw
			Nervonic acid	1.36 ± 0.11 mg/kg dw
[12]	Water/acetone (50:50, *v*/*v*) extract	LC-MS	*Tocopherols*:	
			α-Tocopherol	3.35 ± 0.04 mg/100 g dw *
			β/γ-Tocopherols	1.73 ± 0.01 mg/100 g dw
			δ-Tocopherol	1.47 ± 0.02 mg/100 g dw

GC/MS—gas chromatography–mass spectrometry; LC-MS—liquid chromatography with mass spectrometry; NQ—non-quantified; UTE—ultra-turrax extraction; WSE—walnut septum extract. * The amounts are presented as means ± SD (standard deviation); dw—dry weight.

**Table 4 antioxidants-12-00604-t004:** Carbohydrate compounds identified and quantified in walnut septum.

Ref.	Extracts	Analytical Method	Compounds	Amount
[21]	Petroleum ether, acetone, ethanol extracts	HPGFC-RID	*Monosaccharides:* Arabinose, Galactose, Glucose, Xylose, Mannose	NQ
[25]	WS extract [44]	HPLC-MS	Monosaccharides, disaccharides:	
			Mannose	11.45 ± 0.52 mg/g dw *
			Rhamnose	8.99 ± 0.41 mg/g dw *
			Ribose	2.99 ± 0.27 mg/g dw *
			Glucuronic acid	3.09 ± 0.09 mg/g dw *
			Trehalose	223.76 ± 10.01 mg/g dw *
			Galacturonic acid	8.18 ± 0.05 mg/g dw *
			Xylose	44.79 ± 2.08 mg/g dw *
			Galactose	2.77 ± 0.02 mg/g dw *
			Arabinose	8.11 ± 1.57 mg/g dw *

HPGFC-RID—high-performance gel filtration chromatography coupled with a refractive index detector and an empower workstation; HPLC-MS—high-performance liquid chromatography with mass spectrometry. NQ—non-quantified; * The amounts of are presented as means ± SD (standard deviation); dw—dry weight walnut septum.

**Table 5 antioxidants-12-00604-t005:** Other compounds identified and quantified in walnut septum extracts.

Classification	Compounds	Amount	Ref.
Sesquiterpenoids	Megastigmanes: Blumenol B, Dehydrovomifoliol, (6R,9R)-9-Hydroxymegastigman-4-en-3-one (blumenol C glucoside), (6R,9S)-9- Hydroxymegastigman-4-en-3-one (blumenol C glucoside)	NQ	[22]
	Carotenoids: Dihydrophaseic acid	NQ	
	*Megastigmanes:* Blumenol B	NQ	[25]
	*Megastigmanes:* Diamegastigmane A, Diamegastigmane B, Diamegastigmane C, Blumenol A (vomifoliol), Blumenol B, Aglycone of euodionoside G, Bridelionol C, Myrsinionoside A, Byzantionoside B, (6R, 9S)-60-(400-hydroxybenzoyl)-roseoside	NQ	[38]
	*Taraxasteranes*: Juglansin A, Juglansin B, Juglansin C, Juglansin D	NQ	[39]
Quinones	(4s)-4-hydroxy-1- tetralone	NA	[22]
Coumarins	7-hydroxy-methylcoumarin	240.45–332.18 µg/g dw *	[32]
Amino acids	Lysine (Lys)	2.41 ± 0.02 mg/g dw	[25]
	Phenylalanine (Phe)	1.23 ± 0.02 mg/g dw	
	Threonine (Thr)	1.04 ± 0.03 mg/g dw	
	Isoleucine (Ile)	1.25 ± 0.04 mg/g dw	
	Leucine (Leu)	2.09 ± 0.07 mg/g dw	
	Valine (Val)	1.66 ± 0.04 mg/g dw	
	Aspartic acid (Asp)	0.59 ± 0.03 mg/g dw	
	Serine (Ser)	1.61 ± 0.03 mg/g dw	
	Glutamate (Glu)	3.86 ± 0.02 mg/g dw	
	Glycine (Gly)	2.10 ± 0.05 mg/g dw	
	Alanine (Ala)	1.56 ± 0.02 mg/g dw	
	Cystine (Cys)	2.15 ± 0.04 mg/g dw	
	Tyrosine (Tyr)	0.88 ± 0.04 mg/g dw	
	Proline (Pro)	2.40 ± 0.02 mg/g dw	
	Arginine (Arg)	1.13 ± 0.03 mg/g dw	
	Histidine (His)	2.61 ± 0.04 mg/g dw	
	Methionine (Met)	0.98 ± 0.04 mg/g dw	
Minerals	K	8362.87 ± 21.90 mg/kg dw	[25]
	Na	479.17 ± 6.29 mg/kg dw	
	Ca	3526.37 ± 15.71 mg/kg dw	
	Mg	636.80 ± 12.94 mg/kg dw	
	Fe	36.93 ± 0.90 mg/kg dw	
	Cu	3.78 ± 0.16 mg/kg dw	
	Zn	4.93 ± 0.11 mg/kg dw	
	Mn	14.70 ± 0.10 mg/kg dw	
	Se	0.049 ± 0.002 mg/kg dw	

BPC—bound phenolic compounds; CCC—counter-current chromatography; CD—circular dichroism (spectroscopy); ESI—electrospray ionization; ESI-MS—electrospray ionization mass spectrometry; FPC—free phenolic compounds; HPLC—high performance liquid chromatography; HRESIMS—high resolution electrospray ionization mass spectroscopy; NA—not available; NQ—non-quantified; NMR—nuclear magnetic resonance; UV—ultraviolet spectroscopy; WS—walnut septum/septa; WSE—walnut septum extract; * The concentration range for the three types of walnut septum extracts (EE, UE, ME); The amounts of amino acids and minerals are presented as means ± SD (standard deviation); dw—dry weight walnut septum.

**Table 6 antioxidants-12-00604-t006:** Biological activities reported on in vivo studies performed with walnut septum extracts or compounds isolated from walnut septum.

**Ref.**	**Animals**	**Extract/Substance Tested**	**Groups and Doses**	**Parameters/Biomarkers Analyzed**	**Effects/Biological Activities**
[16]	- Male adult Wistar rats and diabetic male adult Wistar rats (200–250 g) (*n* = 7 animals in each group) - Experimental model of diabetes: streptozotocin-induced (60 mg/kg bw, i.p.)	- WSE preparation: dried and ground septum extracted with 90% ethanol in a Soxhlet apparatus for 48 h; the solvent was evaporated, and the solid extract was dried and suspended in water	- Control group (normo-glycemic rats): distilled water - treated groups (normo-glycemic rats): WSE in different doses (0.1, 0.2, 0.5, 1 g/kg bw/day) by gavage for 14 days - diabetic control group: distilled water - Diabetic treated groups: WSE in different doses (0.1, 0.2, 0.5, 1 g/kg bw/day) by gavage for 14 days	Biochemical parameters: serum glucose, ALAT, ASAT, ALP, insulin levels	- ↓ glycemia (*p* < 0.01 for 0.2 and 1 g/kg bw WSEs, and *p* < 0.001 for 0.5 g/kg bw WSE) in diabetic rats
[14]	- Diabetic male bulb/C mice (25–30 g) divided into five groups - Experimental model of diabetes: streptozotocin-induced (220 mg/kg bw, i.p.)	- WSE preparation: dried and ground septum mixed with water in a blender for 24 h; the solvent was evaporated, and the solid extract was dried and suspended in normal saline	- Control group: normal saline (*n* = 6) - Treated groups: aqueous WSE in different doses (200, 400, 600, or 800 mg/kg bw/day) by gavage for 28 days (*n* = 7–10)	Histopathological analyses of the pancreatic tissues and blood glucose levels	- ↓ glycemia—time-dependent effect (*p* < 0.05) observed for all tested doses - WSE did not change the pancreatic structure
[18]	- Male Wistar rats (180–220 g) - Experimental model of diabetes: alloxan-induced (a single dose 150 mg/kg, i.p.)	- WSE preparation: macerated (1:10 *w*/*v*) in 80% ethanol at room temperature for 2 days; the solvent was evaporated at 40 °C - TPC of dried extract—21.64 ± 1.44 mg * GAE/g	- Control group(healthy): 0.9% saline (*n* = 8) - Diabetic control group: 0.9% saline (*n* = 8) - Positive diabetic control group: metformin (50 mg/kg bw) (*n* = 8) - Treated groups: WSE (200 and 400 mg/kg bw/day, respectively) by gavage, 28 days (*n* = 8)	Antidiabetic and hypolipidemic effects: - Fasting blood glucose on 1st, 14th, and 29th day - Oral glucose tolerance test (OGTT) on 29th day - Lipid profile: TG, TC, LDL-C, HDL-C - Liver enzymes: ALAT, ASAT	Both doses of WSE: - ↓ glycemia (*p* < 0.001) on 14th and 29th days - hypoglycemic effect appeared at 90 min post-treatment - ↓ ASAT (*p* < 0.05) and ALAT (*p* < 0.001) - ↓ TG (*p* < 0.05) Only at the dose of 200 mg/kg: - ↓ TC and LDL-C (*p* < 0.001)
[17]	- White mice (25–27 g), 3 groups - Experimental model of leukopenia	WSE preparation: in water (water:walnut septa, 2:1) and boiled on a water bath until the evaporation of ½ of the water volume	- Control group: intact animals - Group I—cyclophosphamide (350 mg/kg, i.p.) - Group II—WSE (0.2 mL, orally) twice a day after cy- clophosphamide (350 mg/kg, i.p.).	Leucocytes in peripheral blood Peripheral blood cells in blood and bone marrow on the 8th day after the cyclophosphamide administration	- Fast ↑ the number of immature and mature neutrophil in the peripheral blood - WSE stimulated the division, differentiation, and maturation of blast cells (myeloid as well as lymphoid lines) in the bone marrow of mice with leukopenia
[13]	- White mice (25–27 g), 5 groups (*n* = 10)	- WSE preparation in water (water:walnut septa, 2:1) and boiled on a water bath until the evaporation of ½ of the water volume	- Control group: intact animals. - Group I: animals on the 4th day after single i.p. injection of cyclophosphamide (350 mg/kg) - Group II: animals on the 8th day after single i.p. injection of cyclophosphamide (350 mg/kg) - Group III: animals treated orally with WSE (0.2 mL) twice a day after single injection of cyclophosphamide on the day 4 - Group IV: animals treated orally with WSE (0.2 mL) twice a day after single injection of cyclophosphamide on day 8	Sorption capacity of erythrocytes Resistance to hemolysis of erythrocytes	- WSE stimulated the sorption capacity of erythrocytes on the 8th day after the administration of cyclophosphamide - WSE prevented the process of hemolysis, reflected both by ↓ percentage of lysed erythrocytes and lysis period, effect observed both on the 4th and 8th days of application
[13]	- Female Wistar rats (*n* = 21), 9–12 weeks old (200 ± 30 g *), randomly and equally divided into seven groups	- Dried septa powder (2 Kg) extracted with 95% methanol (7 L) by maceration for 7 days at room temperature, concentrated at 40 °C, and lyophilized (JRSME)	- Control group: 0.9% NaCl - Groups 2–7: 10, 100, 1000, 1600, 2900, and 5000 mg/kg bw JRSME suspended in 0.9% NaCl; administered intragastric in a single dose; the animals were monitored 14 days	Acute toxicity assay: lethality, signs of toxicity, and weight daily monitored	- LD50 was estimated to be > 5000 mg/kg bw - any behavioral and macroscopic morphological changes was observed during the experimental period compared to the negative control group
[19]	- Female Wistar rats (*n* = 12), 9–12 weeks old (200 ± 30 g *), randomly and equally divided into two groups	- Dried septa powder (2 Kg) extracted with 95% methanol (7 L) by maceration for 7 days at room temperature, concentrated at 40 °C, and lyophilized (JRSME)	- Group I (control group): 0.9% NaCl - Group II: 1000 mg/kg bw JRSME suspended in 0.9% NaCl; administered by gavage, 1x/day for 4 weeks	Subchronic toxicity assay Biochemical parameters (Cr, ALAT, ASAT, and urea) Antioxidant status: enzyme activities (GPx, SOD, PON-1, AO, XDH) and MDA level in liver and kidney tissues Histological analysis of the liver, kidney, heart, brain, and eye tissue samples Quantitative assessment of kidney tissue changes: epithelium thickness and diameter of renal glomerulus	JRSME: - did not change the blood SOD, GPx, and serum PON-1 activities vs. control group - ↑ serum ALAT (*p* < 0.01) and ASAT (*p* < 0.05) activities after 4 weeks vs. the first week - ↓ blood urea (*p* < 0.05) vs. control group - ↑ serum MDA (*p* < 0.01) vs. control group - ↓ XDH (*p* < 0.001) and AO (*p* < 0.05) in kidney vs. control group - mild histopathological changes in liver and kidney; no structural and pathological changes in heart, brain, and eye tissues
[29]	- Male ICR mice divided into six groups (*n* = 10) - Experimental model of diabetes: streptozotocin-induced (45 mg/kg bw, i.p.)	- Microwave assisted extraction (400 W) with distilled water for a ratio of 20 mL water/g raw material, for 40 min. - Low molecular weight polysaccharide fraction (DJP-2) was separated and purified, then lyophilized	- Control group (healthy): distilled water - Control group (diabetic): distilled water - Positive control group (PG): - 200 mg/kg bw/day metformin (intragastrically, for 10 days). - treated groups (DJP50, DJP100, DJP200): 50 mg/kg bw/day, 100 mg/kg bw/day, and 200 mg/kg bw/day DJP-2 (intragastrically, for 10 days)	Blood glucose levels	DJP-2: - ↓ glycemia (*p* < 0.05) in a treatment dose- and time-dependent manner
[31]	- Healthy nulliparous female Wistar rats (*n* = 32), 3 months old (197.87 ± 15.88 g *), randomly and equally divided into four groups	- Walnut septum (0.5 g) extracted with 5 mL water/acetone (50:50, *v*/*v*) by UTE, followed by complete evaporation of the acetone	- Group 1 (CY): 0.9 % NaCl and SF - Group 2 (D-gal): D-gal and SF - Group 3 (D-gal + WK): D-gal and WK added to SF - Group 4 (D-gal + WSE): D-gal and WSE added to SF D-gal dose: 400 mg/kg bw/day, administered s.c. 3x/week (MWF), 8 weeks, dissolved in 0.9 % NaCl (0.5 mL D-gal solution/100 g bw) WK dose: calculated for 9% of the daily diet given to the animals (quantity/rat); this dose is equivalent to 43 g or 1.5 oz of nuts/2000 kcal/day; administered daily added to SF WSE dose: the equivalent amount of TPC as the WK dose (9%) determined by FC assay; administered daily added to SF	Hematological and biochemical analyses (fasting glycemia, blood lipid profile, ASAT, ALAT, creatinine, urea, CRP) Antioxidant (antiradical) activity (AA) of the brain and/or liver homogenates (by TEAC, DPPH, and FRAP assays) Oxidative stress biomarkers in the brain and liver homogenates: ROS (by DCFH-DA assay), AGEs (by fluorimetry), total NO (by Griess assay), total MDA by UPLC-PDA), total GSH level (by Ellman’s assay) AChE activity (by Ellman’s assay) Histopathological and immunohistochemical analyses of the brain and liver tissues	Both WK and WSE intake normalized some hematological parameters significantly lowered by D-gal (vs. D-gal): - ↑ WBC (*p* < 0.01 for WK and *p* < 0.05 for WSE) - ↑ LY (only WSE, *p* < 0.01) - ↑ HGB (both WK and WSE, *p* < 0.01) WK and WSE intake significantly improved some biochemical parameters affected by D-gal (vs. D-gal): - ↓ glycemia (*p* < 0.05 for WK and *p* < 0.01 for WSE) - ↓ ASAT (*p* < 0.01 for WK and *p* < 0.05 for WSE) - ↓ ALAT (*p* < 0.01 for both WK and WSE) WK or WSE intake significantly improved cellular AA affected by D-gal (*vs.* D-gal): - ↑ TEAC-AA both in liver and brain (only WK, *p* < 0.01) - ↑ DPPH-AA in liver (only WK, *p* < 0.05) - ↑ FRAP-AA in liver (only WK, *p* < 0.05) WK and WSE intake significantly decreased oxidative stress biomarkers affected by D-gal (vs. D-gal): - ↓ ROS in liver (*p* < 0.05 for both WK and WSE) and in brain (*p* < 0.05 for WK and *p* < 0.01 for WSE) - ↓ AGEs in liver (*p* < 0.05 for WK and *p* < 0.01 for WSE) and in brain (only WK, *p* < 0.05) - ↓ total NO in liver (*p* < 0.01 for both WK and WSE) and in brain (*p* < 0.05 for WK and *p* < 0.01 for WSE) - ↓ total MDA in liver (*p* < 0.05 for WK and *p* < 0.01 for WSE) - ↑ total GSH in liver (only WK, *p* < 0.05) WK and WSE intake significantly decreased AChE activity affected by D-gal (vs. D-gal): - ↓ AChE in brain (*p* < 0.01 for both WK and WSE) Both WK and WSE intake kept the thickness of the hippocampal CA1 region and the hyperchromic neuron count in this area, significantly affected by D-gal, closer to that of the control group (CY) (Nissl staining). Both WK and WSE intake protected hepatocytes to be affected by D-gal (that induced an atrophy with attenuation of the nuclei and dilatation and hyperemia of sinusoids in the direction of the centrilobular vein (HE-staining) WK intake, but more evidence for WSE intake, prevented the liver glycogen accumulation induced by D-gal exposure (complete absence of the hepatic glycogen stores in D-gal + WSE group)
[31]	- Healthy male Wistar rats (*n* = 24), 20 months old (352.12 ± 37.70 g *), randomly and equally divided into three groups	- Walnut septum (0.5 g) extracted with 5 mL water/acetone (50:50, *v*/*v*) by UTE, followed by complete evaporation of the acetone	- Group 1 (CO): SF - Group 2 (WK): WK added to SF - Group 3 (WSE): WSE added to SF WK dose: calculated for 9% of the daily diet given to the animals (quantity/rat); this dose is equivalent to 43 g or 1.5 oz of nuts/2000 kcal/day; administered daily added to SF WSE dose: the equivalent amount of TPC as the WK dose (9%) determined by FC assay; administered daily added to SF	Hematological and biochemical analyses (fasting glycemia, blood lipid profile, ASAT, ALAT, creatinine, urea, CRP) AA of the brain and/or liver homogenates (by TEAC, DPPH, and FRAP assays) Oxidative stress biomarkers in the brain and liver homogenates: ROS (by DCFH-DA assay), AGEs (by fluorimetry), total NO (by Griess assay), total MDA by UPLC-PDA), total GSH level (by Ellman’s assay) AChE activity (by Ellman’s assay) Histopathological and immunohistochemical analyses of the brain and liver tissues	WK or WSE intake significantly improved some biochemical parameters affected by natural aging (*vs.* CO): - ↓ glycemia (only WSE, *p* < 0.05) - ↓ ASAT (*p* < 0.01 for WK and *p* < 0.05 for WSE - ↓ ALAT (*p* < 0.05 for both WK and WSE) WK or WSE intake significantly improved cellular AA affected by natural aging (*vs.* CO): - ↑ TEAC-AA in liver (only WK, *p* < 0.05) - ↑ TEAC-AA in brain (for both WK and WSE, *p* < 0.05) - ↑ DPPH-AA in liver (only WK, *p* < 0.05) WK and WSE intake significantly decreased of oxidative stress biomarkers affected by natural aging (*vs.* CO): - ↓ ROS in liver (only WK, *p* < 0.05) - ↓ AGEs in liver (*p* < 0.01 for both WK and WSE) - ↓ total MDA in liver (*p* < 0.05 for WSE) Both WK and WSE intake significantly lowered AchE activity in old rats (*vs.* CO): - ↓ AchE in brain (*p* < 0.05 for WK and *p* < 0.01 for WSE) Only WSE intake slightly reduced the thickness of the hippocampal CA1 region and the hyperchromic neuron count *vs* CO (Nissl staining) WK intake reduced glycogen deposits in the hepatic parenchyma; this effect was more highlight for WSE intake
[34]	*Caenorhabditis elegans* in vivo model	2-carboxy-5,7-dihydroxy-3-naphthyl-β-D-glucopyranoside (CDG) was isolated from DJF	- 50, 100, and 200 μg/mL of CDG	Lifespan assay and lipofuscin accumulation Stress resistance assays: the thermotolerance assay and the resistance to paraquat ROS level, antioxidant enzyme activities, and MDA level	- ↑ lifespan of the nematodes treated with 50, 100, and 200 μg/mL CDG increased in a concentration-dependent manner by 15.0%, 21.6%, and 32.1%, respectively - ↓ accumulation of lipofuscin by CDG (*p* < 0.05 vs. control) in a concentration-dependent manner After heat stress treatment, survival rates increased for CDG exposure with 67.34%, 77.89%, and 94.55%, in a dose-dependent manner (vs. 47.85% for control) CDG (200 μg/mL) increased lifespan of nematodes exposed to paraquat by 40.83% relative to the control group CDG (50, 100, and 200 μg/mL) showed antioxidant activity in nematodes: - ↓ ROS levels by 21.56%, 47.75%, and 55.17%, respectively - ↑ SOD activity: 1.29 (*p* < 0.05), 1.38 (*p* < 0.05), and 1.43 times (*p* < 0.01), respectively - ↑ CAT activity: 1.07, 1.43, and 1.76 times, respectively - ↓ MDA levels by 49.8%, 57.5%, and 65.8%, respectively
[35]	- Adult male Kunming mice (18–22 g) divided into five groups (*n* = 12)	- DJE-H extract: DJ powder was extracted with hot water (1:20, *w*/*v*) at 85 °C for 3 h, concentrated at 45 °C under low pressure, and freeze-dried - DJE-A extract: DJ powder was extracted with 70% ethanol at room temperature, concentrated, and freeze-dried	- Normal group (N): vehicle - Control group (C): vehicle - DJE-H group: DJE-H (0.1 mL/10 g bw) - DJE-A group: DJE-A (0.1 mL/10 g bw) - Positive control group (PC): taurine All treatments were intragastrical, for 30 days	The climbing bar time and the force swimming test at 30 min after final treatment Serum LDH activity SUN level Liver and muscle glycogen levels TAC, SOD, GSH, CAT, and MDA in serum, liver, and skeletal muscle	- ↑ bar climbing time by 9.6 %, 22.8 % (*p* < 0.05), and 21.3% in DJE-H, DJE-A, and PC groups, respectively, vs. C group - ↑ forced swimming time by 45.4%, 97.6% (*p* < 0.05), and 47.3% (*p* < 0.05) in DJE-H, DJE-A, and PC groups, respectively, vs. C group - ↑ LDH (*p* < 0.05) in DJE-H, DJE-A, and PC groups, respectively, vs. C group both in blood and muscle tissue - ↓ SUN levels (*p* < 0.05) in DJE-H, DJE-A, and PC groups, respectively, vs. C group - ↑ glycogen levels (*p* < 0.05) in DJE-H, DJE-A, and PC groups, respectively, vs. C group both in liver and muscle tissues - ↓ MDA (*p* < 0.05) in DJE-A and PC groups, respectively, vs. C group, in muscle - ↑ TAC (*p* < 0.05) in DJE-H, DJE-A, and PC groups, respectively, vs. C group, both in serum, liver (without DJE-H) and muscle - ↑ SOD (*p* < 0.05) in DJE-H, DJE-A, and PC groups, respectively, vs. C group, in muscle, and only in DJE-A and PC groups, respectively, vs. C group, in serum - ↑ GSH and CAT (*p* < 0.05) in DJE-H, DJE-A, and PC groups, respectively, vs. C group, both in serum, liver, and muscle
[36]	- Healthy male Wistar rats (*n* = 24), 3 months old (227.75 ± 14.60 g *), randomly and equally divided into four groups	- Walnut septum (0.5 g) extracted with 5 mL water/acetone (50:50, *v*/*v*) by UTE, followed by complete evaporation of the acetone	- Group 1 (NC): distilled water (negative control) - Group 2 (Cd): codeine phosphate (positive control) - Group 3 (WSE) - Group 4 (WSE 1:2) Cd dose: 3 mg /kg bw/day, administered orally by gavage, 1x/day, 3 consecutive days WSE dose: 10 mL WSE (containing 134 mg GAE)/kg bw/day, administered orally by gavage, 1x/day, 3 consecutive days WSE 1:2 dose: 5 mL WSE (containing 134 mg GAE)/kg bw/day, administered orally by gavage, 1x/day, 3 consecutive days [mg GAE were calculated analyzing TPC by FC assay]	The number and frequency of coughs caused by citric acid aerosols (17.5 %; 4 min-period of exposure, 1 h after the last oral treatment; 8 min-period of observation), and cough latency after exposure. Oxidative Stress Biomarkers (6 h after the citric acid aerosol exposure): TAC in serum (by TEAC assay); NO and ROS levels in lung homogenates Inflammatory biomarkers (6 h after the citric acid aerosol exposure): IL-6, CXC-R1, and CXC-R2 Histopathological analysis of lung tissue (6 h after the citric acid aerosol exposure)	Significant antitussive effect of WSE, comparable with codeine: - relative frequency of coughs: *p* < 0.01 *vs* NC (67.50% of inhibition) for WSE, and *p* < 0.01 *vs* NC (51.57 % of inhibition) for Cd - cough latency: *p* < 0.05 *vs* NC for WSE, and *p* < 0.01 *vs* NC for Cd WSE significantly decreased oxidative stress in lung, induced by citric acid aerosol exposure: - ↓ ROS (both WSE and WSE 1:2, *p* < 0.01 *vs* NC and *vs* Cd) WSE showed anti-inflammatory activities in lung tissue, comparable with codeine: - ↓ IL-6 (*p* < 0.01 for both WSE and WSE 1:2, and Cd, *vs* NC) - ↓ CXC-R1 (*p* < 0.05 for both WSE and Cd, *vs* NC) Alveolar space area was the largest for WSE, followed by Cd and WSE 1:2, and the smallest for NC

* Mean ± SD; AA-antioxidant activity; AChE-acetylcholinesterase; AGEs-advanced glycation end products; ALAT-alanine aminotransferase; ALP-alkaline phosphatase; AO-aldehyde oxidase; ASAT- aspartate aminotransferase; bw-body weight; CDG-2-carboxy-5,7-dihydroxy-3-naphthyl-β-D-glucopyranoside; CO-control old; Cr-creatinine; CRP-C-reactive protein; CY-control young; CXC-R1-chemokine receptor type 1; CXC-R2—chemokine receptor type 2; DCFH-DA-2′,7′-dichlorodihydrofluorescein diacetate; D-gal-D-galactose; DPPH-2,2-diphenyl-1-picrylhydrazyl; DJF- Diaphragma juglandis fructus; FC-Folin-Ciocâlteu; FRAP-ferric reducing antioxidant power; GSH-reduced glutathione; GPx-glutathione peroxidase; HDL-C-high density lipoprotein-cholesterol; HE-hematoxylin and eosin; HGB-hemoglobin; IL-6-Interleukin 6; i.p.-intraperitoneal; ISWF-Internal septum of walnut fruit; JRSME-methanolic extract of *Juglans regia* septum; LDH-lactate dehydrogenase; LDL-C-low density lipoprotein-cholesterol; LY- lymphocytes; MDA-malondialdehyde; MWF-Monday, Wednesday, and Friday; NC-negative group; NO-nitric oxide; PON-1-paraoxonase 1; ROS-reactive oxygen species; s.c.-subcutaneously; SF-standard feed; SOD-superoxide dismutase; SUN-serum urea nitrogen; TAC-total antioxidant capacity; TC-total cholesterol; TEAC-Trolox equivalent antioxidant capacity; TG-triglycerides; TPC-total phenolic content; UPLC-PDA-ultraperformance liquid chromatography with photo diode array detector; UTE-ultra-turrax turboextraction; WBC-white blood cells; WK-walnut kernel; WSE-walnut septum extract; XDH-xanthine dehydrogenase.

## Data Availability

Data are contained within the article.

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
