# Peer review of "Phytochemicals and Biological Activities of Walnut Septum: A Systematic Review"

_antioxidants, 2023, doi:10.3390/antiox12030604_

Round 1

Reviewer 1 Report

The work focused on a critical review assessment of the information published regarding walnut septum chemical composition as well as its related biological properties. The review manuscript is very well written.  It tries to concisely summarize the reported biological compounds in walnut septum as well as its biological properties. The first part is very descriptive, but in the second part, the authors are describing but also discussing the reported biological properties with that of other matrices. This long review may be of interest to the scientific community dealing with plant materials, not only walnut by-products. In my opinion, the review manuscript is suitable for publication after minor revision.

Tables can be adapted to reduce their sizes, maybe decrease line spacing.

Line 51: Correct references to [6,11]

Section 2.1. Eligibility criteria. Some information provided in this section should be explained in a different way, for example: (1) languages that were not known (maybe it is better to specify languages that were selected), (2) publications with full text not available (have the authors consider to ask the corresponding authors a copy of their work?). These criteria exclude the concept of exhaustive systematic/comprehensive review employed by the authors.

Line 208: (REF)? Maybe references are missing?

Line 269: Correct references to [33,41]

Conclusions need to be improved from a critical point of view. These conclusions are just a very brief summary of what has been reported in the manuscript. So try to highlight the most outstanding results described (for example, but not only, the most outstanding biological property and why). 

Reviewer 2 Report

The present paper "Phytochemicals and Biological Activities of Walnut Septum: a Systematic Review" aims the review articles regarding the walnut septum chemical composition and the related biological activities, including antioxidant activities, anti-inflammatory effects, antimicrobial properties, antidiabetic activities, anti-tumor properties, and anti-aging potential. The present paper is valuable, but it is necessary to review it in order to simplify the reading process.

First of all, I think, this paper is too long... In my opinion, some information in the tables is very detailed, for example, the conditions for extract preparation, chromatographic conditions, etc. I think, if the reader will be interested in the sample preparation and analysis technique, he will read the original study. In the review paper, it should be presented only the main information and most important findings of other studies.

Unify the column "Study type" in Table 1, as now in some rows "in vitro", other - "phytochemical analysis", and etc.

I think, compounds with "NQ" in the tables, may be listed in a line, not a different row in order to keep space.

Table 1: letters a, b, and c should be in a superscript position in dwa, dwb, and dwc.

Is there an official pharmacopeia or WHO recommendation for the correct name: walnut septum or walnut diaphragm? I should recommend using one term throughout the manuscript, as now even in one table two different names for one object are used.

When talking about the chemical composition, it would be valuable to add not only the quantity of these compounds but also the main biological activity. 

Latin names in the reference list must be in italic.

Check the journal requirements for manuscript preparation, as in my opinion, it should not be any spacing after each paragraph.

Reviewer 3 Report

The presented manuscript is focused on the walnut septum chemical composition and its health properties.  The methodology of this review was described in detail. The paper is well prepared and well organized in general. However, I have a few comments.
In the whole manuscript, the authors should delete the spaces between the paragraphs inside the same chapter
Chapter 3. “Chemical composition” change to “Chemical Composition”
There are many small mistakes in the Tables. For example please check for Table 1 Ellagic acid hexoside – it is an empty place NQ or an inappropriate line?, μg/g or μg/g dw a, Catechin – lack of data or data in an inappropriate line, etc. Include  the spaces between number and % and between number and °C,
 Chapter 3.4 Please describe shortly Table 5 and 6.
Line 547: Include space between the bracket and text.
Conclusions. Such a conclusion can be written for many plant materials. Please enhance what is exceptional in walnut septum compared with others plant materials.

Round 2

Reviewer 3 Report

The Authors corrected the manuscript accordingly.